# A dual role in regulation and toxicity for the disordered N-terminus of the toxin GraT

Ariel Talavera[1,2], Hedvig Tamman[3], Andres Ainelo[3], Albert Konijnenberg[1,2,4], San Hadži[1,5], Frank Sobott[4,6,7], Abel Garcia-Pino [8], Rita Hõrak[3] & Remy Loris[1,2]

Bacterial toxin-antitoxin (TA) modules are tightly regulated to maintain growth in favorable conditions or growth arrest during stress. A typical regulatory strategy involves the antitoxin binding and repressing its own promoter while the toxin often acts as a co-repressor. Here we show that *Pseudomonas putida graTA*-encoded antitoxin GraA and toxin GraT differ from other TA proteins in the sense that not the antitoxin but the toxin possesses a flexible region. GraA auto-represses the *graTA* promoter: two GraA dimers bind cooperatively at opposite sides of the operator sequence. Contrary to other TA modules, GraT is a de-repressor of the *graTA* promoter as its N-terminal disordered segment prevents the binding of the $GraT_2A_2$ complex to the operator. Removal of this region restores operator binding and abrogates GraT toxicity. GraTA represents a TA module where a flexible region in the toxin rather than in the antitoxin controls operon expression and toxin activity.

[1] Structural Biology Brussels, Department of Biotechnology, Vrije Universiteit Brussel, B-1050 Brussel, Belgium. [2] Molecular Recognition Unit, Structural Biology Research Center, Vlaams Instituut voor Biotechnologie, B-1050 Brussel, Belgium. [3] Institute of Molecular and Cell Biology, University of Tartu, 51010 Tartu, Estonia. [4] Biomolecular and Analytical Mass Spectrometry Group, Department of Chemistry, University of Antwerp, Groenenborgerlaan 171, B-2020 Antwerpen, Belgium. [5] Department of Physical Chemistry, Faculty of Chemistry and Chemical Technology, University of Ljubljana, 1000 Ljubljana, Slovenia. [6] Astbury Centre for Structural Molecular Biology, University of Leeds, Leeds LS2 9JT, UK. [7] School of Molecular and Cellular Biology, University of Leeds, Leeds LS2 9JT, UK. [8] Biologie Structurale et Biophysique, Institut de Biologie et de Médecine Moléculaires, Université Libre de Bruxelles, B-6041 Gosselies, Belgium. Correspondence and requests for materials should be addressed to A.T. (email: atalaver@ulb.ac.be) or to R.L. (email: remy.loris@vub.be)

Toxin–antitoxin (TA) modules, the small genetic elements believed to be involved in prokaryotic stress response[1–3], are widespread among both archaea and bacteria[4,5]. Six major types (I–VI) of TA systems have been discovered so far[6]. In each type, the toxin is a protein that interferes with vital cellular processes, but the nature and mode of action of the antitoxin varies. The antitoxin either prevents production of the cognate toxin as an antisense RNA (type I) or as an RNase that degrades the mRNA encoding the toxin (type V), counteracts the activity of the toxin as a protein (type II) or RNA species (type III) that binds to the toxin, acts as an antagonist for the toxin by competing with its target (type IV) or functions as a proteolytic adaptor that promotes degradation of the toxin (type VI). Of these, type II systems with protein antitoxins are the most abundant and widely researched[1].

The production of type II TA proteins is auto-regulated at the level of transcription. Their antitoxins are typically composed of two domains: a DNA binding domain next to an (often intrinsically disordered) toxin neutralizing domain[7]. The DNA binding domain interacts with the operator to inhibit transcription of the TA operon. For many type II TA systems (e.g., *phd/doc*, *ccdAB*, *relBE* and *kis/kid*) repression depends on the ratio between toxin and antitoxin by a mechanism known as conditional cooperativity[8–10]. At low toxin/antitoxin molar ratios, the toxin enhances the antitoxin gene repression, but when a certain threshold ratio is surpassed, the toxin becomes a derepressor. This behavior is generated in different TA modules via distinct molecular mechanisms[11–14]. Nevertheless, other type II TA systems such as *Escherichia coli dinJ/yafQ*, *Proteus vulgaris higBA* or *E. coli mqsRA* are not regulated by conditional cooperativity[15–17]. In the first two cases the toxin does not affect binding of the antitoxin to the operator. For the latter, the toxin disrupts the antitoxin-operator complex.

GraTA (Growth rate affecting Toxin–Antitoxin) is a type II TA module recently discovered in *Pseudomonas putida*[18]. By sequence similarity, GraTA is most closely related to the *higBA* TA family[18]. The toxin GraT has a very mild effect at the optimal growth temperature of 30 °C or higher and this allows for the deletion of the antitoxin *graA* gene from the chromosome. At lower temperatures, however, GraT causes severe growth repression[18]. GraT inhibits ribosome biogenesis and causes the accumulation of nearly complete yet immature ribosome subunits[19]. The antitoxin GraA binds to the *graTA* promoter and effectively represses transcription of the operon[18]. GraA is an unusually stable protein in comparison to most TA antitoxins with a minimal observed half-life of 1 h in cell lysate. It is not degraded by either Lon or Clp that commonly target antitoxins. Instead, its degradation is initiated by an unidentified endoprotease[20]. These properties of the antitoxin result in very efficient inhibition of GraT even when the toxin is ectopically overexpressed[18].

Most type II antitoxins contain an intrinsically disordered region that is required not only for neutralizing the toxin and forming the repressor complex[7,21] but also for its rapid degradation[22] and for the dissociation of the toxin from its target (e.g., CcdB and Gyrase)[7,13]. On the contrary, all toxins characterized to date are fully folded proteins[7]. Here we show that GraA does not contain unstructured regions and forms a globular dimer while the toxin GraT contains an N-terminal intrinsically disordered region that is key for transcriptional regulation of the *graTA* operon as well as for the RNase activity of GraT. GraA binds tightly to the *graTA* operator and GraT prevents this interaction through steric interference from its N-terminal disordered region. Removal of this region restores operator binding, and also abrogates GraT toxicity. GraTA thus represents a type of TA module where intrinsically disordered region in the toxin rather than in the antitoxin controls both operon expression and toxin activity.

## Results

**GraA is a fully folded antitoxin.** The crystal structure of GraA was determined at 1.96 Å (Fig. 1a and Supplementary Table 1). The protein was completely traced and, in contrast to other antitoxins, does not contain an intrinsically unfolded domain. GraA forms a homo-dimer (from now on referred to as GraA$_2$). The GraA monomer consists of one long and four short (2–3 turns) α-helices, of which helices α2 and α3 form a helix-turn-helix (HTH) motif (Fig. 1a). Helix α5 extends over seven turns to form a dimerization unit and its C-terminus (after a kink) also caps over the globular domain of the other monomer.

The fold of GraA is similar to that of the HigA antitoxins from *E. coli* (Protein Data Bank (PDB) codes, 2ICT and 2ICP)[23], *Coxiella burnetti* (PDB code 3TRB)[24] and *P. vulgaris* (PDB code,

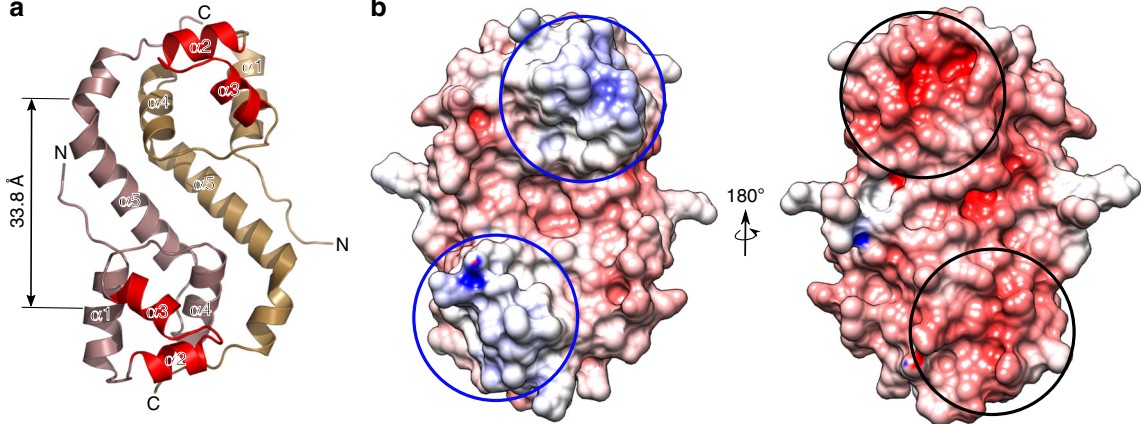

**Fig. 1** Structure of the GraA dimer. **a**. Cartoon representation of GraA$_2$, one monomer is colored sand and the other is in purple. The five α-helices are labeled from α1 to α5. N- and C-termini are labeled. The helix-turn-helix (HTH) motifs are highlighted in red, and the distance between both motifs, measured as the distance between the centers of mass of both α3 helices, is indicated. **b** Van der Waals surface of the GraA dimer colored according to its electrostatic surface potential (10 kT blue, 0 white, −10 kT red) in two orientations rotated by 180°. The orientation on the left corresponds to the same orientation as in (**a**). The DNA binding surface shows a prominent positive electrostatic potential (indicated by a blue circle) while the GraT binding site, located at the opposite side of the GraA$_2$ dimer, is negatively charged (indicated by a black circle)

4MCX[16]. The closest homolog with known structure for GraA is *C. burnetti* HigA (backbone root-mean-square deviation (RMSD) of 2.6 Å for residues 4–99 when superimposing monomers). For *P. vulgaris* HigA, which is characterized in more detail and with which it shares 27% sequence identity, the backbone RMSD is 3.0 Å for residues 4–99 (Supplementary Figure 1A, B).

The dimerization interface of $GraA_2$ is mainly stabilized by hydrophobic interactions. The helix-turn-helix DNA binding motif, comprising residues from 25 to 45, presents a positively charged surface (Fig. 1b). The separation between the two HTH motifs in the dimer, measured as the distance between the centers of mass of both α3 helices, is 33.8 Å (Fig. 1a). Other antitoxins with known structure and bearing HTH DNA binding motifs are the three HigAs mentioned above, *E. coli* MqsA[25], and the HipBs from *E. coli*[26] and *Shewanella oneidensis*[27] (PDB entries 3O9X, 3DNV and 4PU3, respectively). From these antitoxins only *E. coli* HigA has a similar separation between the two HTH motifs of 35 Å, while in the HipB and MqsA dimers the HTH motifs are much closer to each other (~26 Å).

**Two $GraA_2$ bind simultaneously on the operator DNA.** GraA represses the *graTA* operon by binding to its operator[18]. We determined the crystal structure of $GraA_2$ in complex with the full 33 bp operator at 3.8 Å resolution (Supplementary Table 1). Surprisingly, in the crystal two non-interacting GraA dimers bind at opposite sides of the double helix (Fig. 2a, Supplementary Figure 2A, B). The total buried surface area between the two GraA dimers and the DNA is 2234 Å². This interface is evenly distributed among the four partial interfaces formed between each of the four GraA monomers and the DNA strands, with an average value of 558 Å² per GraA monomer.

$GraA_2$ reads the operator in a very particular way. Each $GraA_2$ has one of its HTH motifs interacting with the central TAACGTTA palindrome (Fig. 2b). This "central binding" HTH motif binds with its interaction helix α3 to a TAAC half palindrome. This interaction involves Pro39 protruding into the major groove of the DNA, while Arg46 and Asn42 make hydrogen bonds with the DNA backbone (Fig. 2c). No hydrogen bonds between the protein and the DNA bases are observed. This central binding mode is identical for each of the two bound dimers.

The other HTH motifs of each $GraA_2$ act as the "support" subunits and interact with two distal half sites with sequences TAAG and TAAC, respectively (Fig. 2b). In these distal half sites the interaction is again directed towards the DNA phosphate backbone, and no base-specific contacts are made between protein and DNA.

The operator binding does not significantly influence the structure of $GraA_2$. A monomer backbone RMSD of 1.2 Å is observed between the bound and unbound states of GraA dimer. Binding does not affect the conformation of the GraA monomer nor the relative position or orientation of the monomers in the dimer. The DNA structure also does not deviate much from the ideal B-DNA conformation and no significant bending is observed. In contrast, the shorter distances between the binding helices in HipB and MqsA induce bends in the DNA of 55° and 70°, respectively[25–27] (Supplementary Figure 2C, D).

**GraA interactions with the distal sites are non-specific.** Isothermal titration calorimetry (ITC) experiments between $GraA_2$ and the 33 bp operator fragment point towards a 2:1 stoichiometry in agreement with the crystal structure and indicate a macroscopic binding constant of $3 \times 10^{13}\,M^{-2}$ (Table 1 and Supplementary Figure 3A). To further understand the specificity of $GraA_2$ to its operator, we performed a series of ITC measurements titrating different mutant variants of the operator region into $GraA_2$ (Table 1 and Supplementary Figure 3). $GraA_2$ displays $10^2$ to $10^4$ times lower affinities for operators with mutated palindrome sequences TAACGggc, gccCGTTA and TAgtGTTA (Table 1 and Supplementary Figure 3B, C, D). This confirms the central palindrome as a major binding motif within the operator. However, $GraA_2$ was insensitive (<10-fold decrease in affinity) to all tested operator variants with substitutions in the secondary binding sites outside the central palindrome (Table 1 and Supplementary Figure 3E, F, G). Interestingly, making the operator symmetric based on either the sequence of the left or right side of the central palindrome reduces the affinity for GraA 10-fold (Table 1 and Supplementary Figure 3H, I). We thus conclude that despite the absence of base-specific hydrogen bonds between protein and DNA, the interactions between $GraA_2$ and the operator central palindrome is sequence specific. In contrast, the interactions of the second HTH motif of each $GraA_2$ dimer

| Table 1 Thermodynamics of operator binding | | | |
|---|---|---|---|
| Operator variant | Sequence[a,b] | $K_A$ (M$^{-2}$) | $\Delta H$ (kcal/mol) |
| Wild type | AAATTAACGAA**TAACGTTA**AGCATTCAGCTCAT | $3.0*10^{13} \pm 3.1*10^6$ | −19.0 ± 1.5 |
| Variant 1 | AAATTAACGAA**TAACG**ggccGCATTCAGCT | $2.1*10^9 \pm 5.2*10^6$ | N.D.[c] |
| Variant 2 | AAATTAACGAA**gccCGTTA**AGCcggCAGCT | $4.7*10^{11} \pm 1.4*10^5$ | −5.2 ± 0.6 |
| Variant 3 | AAATTAACGAA**TAgtGTTA**AGCATTCAGCT | $3.9*10^{11} \pm 7.0*10^4$ | −5.6 ± 0.3 |
| Variant 4 | AAAggcACGAA**TAACGTTA**AGCATTCAGCT | $4.3*10^{13} \pm 3.5*10^5$ | −14.6 ± 0.5 |
| Variant 5 | AAATTAACGccg**AACGTTA**AGCATTCAGC | $7.1*10^{12} \pm 4.5*10^5$ | −15.4 ± 0.8 |
| Variant 6 | AAATTAACGAA**TAACGTTA**AGCcggCAGCT | $1.5*10^{13} \pm 5.4*10^5$ | −14.8 ± 0.9 |
| Left side mirrored | AAATTAACGAA**TAACGTTA**ttcgttaattt | $5.2*10^{12} \pm 5.5*10^5$ | −23.4 ± 1.5 |
| Right side mirrored | agctgaatgct**TAACGTTA**AGCATTCAGCT | $3.5*10^{12} \pm 7.7*10^5$ | −26.7 ± 2.2 |
| Central palindrome | GAA**TAACGTTA**AG | N.D. | N.D. |
| Upstream half | AAATTAACGAA**TAACGTTA**AG | N.D. | N.D. |
| Downstream half | AA**TAACGTTA**AGCATTCAGCT | N.D. | N.D. |

[a]Mutations relative to the wild-type sequence are given in small letters
[b]Position of the Palindromic sequence is marked in bold
[c]Not determined (N.D.) because binding was too weak

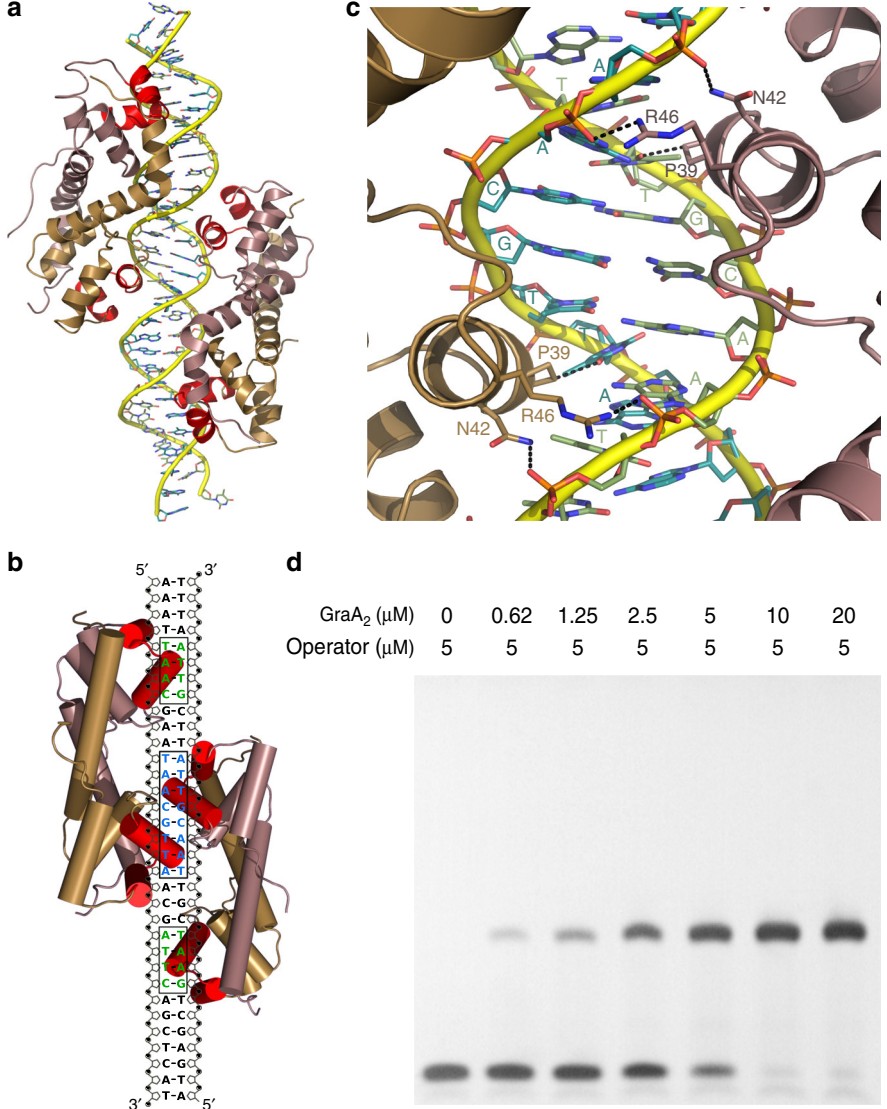

**Fig. 2** GraA$_2$ binds to its operator with a 2:1 stoichiometry. **a** Cartoon representation of the GraA$_2$-DNA-GraA$_2$ complex with the GraA monomers colored as in Fig. 1a. Two GraA dimers are bound at opposite sides of the DNA duplex without making physical contact to each other. **b** Schematic representation of GraA-DNA binding. The central palindrome that provides the two specific half sites is shown in blue and boxed. The two half sites on the DNA that provide non-specific but necessary contacts are colored green. **c** Zoom-in to the central GraA$_2$-DNA-GraA$_2$ binding interface. Amino acids that make direct contact with the DNA are shown in stick and labeled. Hydrogen bonds are shown as dashed lines. **d** Electrophoretic mobility shift assay (EMSA) experiment titrating GraA on the same 33 bp DNA fragment as used for crystallography. Only a single species of protein-DNA complex is apparent. The DNA is saturated in a 2:1 molar ratio of GraA$_2$ to DNA

with the distal binding sites only contribute weakly if at all to specificity.

**Operator binding by GraA is highly cooperative**. Electrophoretic mobility shift assays (EMSAs) of the 33 bp operator fragment with GraA$_2$ confirm the 2:1 (GraA$_2$/operator) stoichiometry, but fail to reveal a two-step binding process as only a single band for the operator complex can be detected (Fig. 2d). This suggests that binding of two GraA$_2$ dimers to its operator is highly cooperative and occurs via a tri-molecular reaction. To evaluate this hypothesis, we performed a series of analytical size exclusion chromatography (SEC) experiments using different GraA$_2$/DNA ratios (Fig. 3a). In all SEC runs, independent of the stoichiometry of the initial mixture, only a single species of GraA$_2$/DNA complex was observed. This species has an apparent molecular weight of about 70 kDa, consistent with two GraA

dimers binding coincidently to the operator ((GraA$_2$)$_2$/DNA). No intermediate species with GraA$_2$/DNA stoichiometry is detected. These results were subsequently confirmed by similar titrations followed via native mass spectrometry. Again, free DNA, free GraA$_2$ and the (GraA$_2$)$_2$/DNA complex were the only species that could be identified (Fig. 3b, Supplementary Figure 4).

To further quantify the apparent cooperativity in the GraA-operator interaction, we performed ITC titrations with shorter versions of the operator comprising only the central palindrome or the palindrome plus its "upstream half-site" or its "downstream half-site" (Table 1 and Supplementary Figure 3J, K, L). These experiments show that the upstream and downstream half sites are both essential for operator binding and that a single GraA dimer will not, on its own, interact with an operator fragment even if the full sequence that is recognized by a single GraA dimer is present.

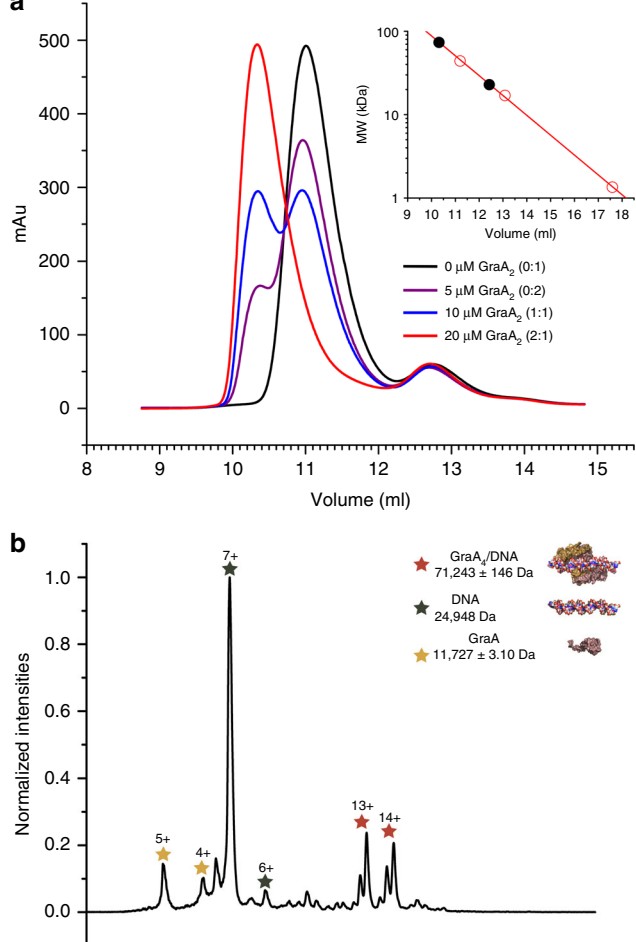

**Fig. 3** GraA$_2$ binding to the operator is highly cooperative. **a** Analytical size exclusion profiles for different GraA$_2$:operator molar ratios using a constant DNA concentration of 10 μM, in a 70 kDa cut-off Bio-Rad SEC 70 column. The inset shows the column calibration curve using molecular weight standards (ovalbumin 44 kDa, myoglobin 17 kDa, vitamin B12 1.35 kDa, open red circles on inset) together with the elution positions for GraA$_2$ and (GraA$_2$)$_2$-operator complex (black dots on inset). **b** Native mass spectrum for a mixture of operator (20 μM of duplex) and GraA$_2$ (20 μM of dimer). The observed species correspond to the operator duplex, GraA$_2$ and (GraA$_2$)$_2$-operator complex

While the two GraA dimers come close together on the operator, they do not effectively touch each other, and oligomerization of GraA$_2$ therefore can be excluded as a driving force for the apparent obligatory co-incident binding. Cooperativity is therefore mediated entirely via the common DNA ligand, although the exact mechanism remains obscure.

**GraTA architecture deviates from the *P. vulgaris* HigBA**. We determined the structure of the GraTA toxin–antitoxin complex at 2.2 Å resolution (Supplementary Table 1, Fig. 4a). With one His-tagged GraT monomer bound to each GraA monomer, a GraT-GraA$_2$-GraT hetero-tetramer (hisGraT$_2$A$_2$ for short) is formed. The structure of GraT is similar to *P. vulgaris* HigB (PDB entry 4MCX) with a backbone RMSD of 2.5 Å for residues 23–92 and a sequence identity of 29% (Supplementary Figure 1B, C).

Binding of GraT does not affect the structure of the GraA monomer but the relative orientation of each monomer in the GraA dimer is somewhat displaced. Upon binding of GraT, there is a 2.2° relative rotation of the GraA helix α3, which results in a 1.6 Å decrease of the distance between the HTH motifs (Supplementary Figure 5A–D).

GraT binds on a negatively charged surface area of GraA, which is located opposite to the DNA binding surface (Fig. 1b). The contact surface area of one GraT monomer with GraA$_2$ is 488 Å². In GraT, residues S27, E30, R31, K32, A38, R43, D44, S47 and G50 are involved in hydrogen bonds with the GraA residues R7, E13, E18, D64, T66 and N72 (Fig. 4b). Hydrophobic interactions are also important for the GraA-GraT binding. In particular, A34, M35, A38, A39, P48 and P49 of GraT interact with a hydrophobic pocket on the surface of GraA formed by I9, I14, F19, M23, L63, F69 and L73.

Based on extrapolation from the structure of HigB in complex with the ribosome[28] (PDB entry 4W4G), the active site of GraT, formed by a cleft around His92, is not occluded by GraA. Instead, GraA inactivates GraT by inhibiting its binding to the ribosome due to steric hindrance via clashes between GraA and the ribosomal RNA as well as ribosomal protein S13 (Supplementary Figure 5E).

Despite their structural similarities, the relative orientations of the toxin and antitoxin in the GraTA complex are different from what is seen in the *P. vulgaris* HigBA complex[16]. When both antitoxins are superimposed, the bound toxins GraT and HigB show a relative rotation of about 25° and their centers of mass are displaced by 6.8 Å (Fig. 4c). Still, about 30% of the GraA residues involved in the GraA–GraT interface are conserved in *P. vulgaris* HigA (Supplementary Figure 1B). In contrast, none of interface residues of GraT are conserved in *P. vulgaris* HigB (Supplementary Figure 1B).

**The N-terminus of GraT is intrinsically disordered**. Another major difference between the *graTA* and the *P. vulgaris higBA* modules concerns the N-termini of the toxins. In *P. vulgaris* HigB the N-terminus is well defined, both in the free state of HigB[28] and when bound to HigA[16] (PDB entries 4PX9 and 4MCX, respectively), and also when bound to the ribosome (PDB entries 4YPB and 4W4G)[29]. In GraT, on the other hand, the first 22 N-terminal residues together with the 6-His tag are located in an interstitial space and could not be traced due to lack of electron density. These 22 residues do not correspond to an extension that is absent in *P. vulgaris* HigB, but to a stretch that is common to both proteins (Supplementary Figure 1B). In *P. vulgaris* HigB, this region folds into a strand-helix-loop structure. Of this fold, the N-terminal β-strand is conserved in *E. coli* RelE but the short α-helix is present in a different orientation[12].

In order to validate that the intrinsically disordered N-terminus of GraT is indeed retained in solution and not an artifact of crystal packing or His-tag placement, we designed a tag-free construct of the GraT$_2$A$_2$ complex that was further analyzed by small-angle X-ray scattering (SAXS). To obtain the tag-free GraT$_2$A$_2$ complex, a TEV (tobacco etch virus) protease cleavage site was inserted in between the 6-His tag and the second residue of GraT (ΔHisGraT$_2$A$_2$). The tag-free GraT$_2$A$_2$ complex only differs from wild-type GraT$_2$A$_2$ by having a glycine instead of a methionine as the first residue of GraT.

At first glance, the dimensionless Kratky plot of GraT$_2$A$_2$ shows that the complex is globular (Fig. 5a). Nevertheless, as the predicted disorder part is only 11% of the full complex and might not be captured by the SAXS experiment, we wanted to know which possible conformation of GraT$_2$A$_2$ best fits the

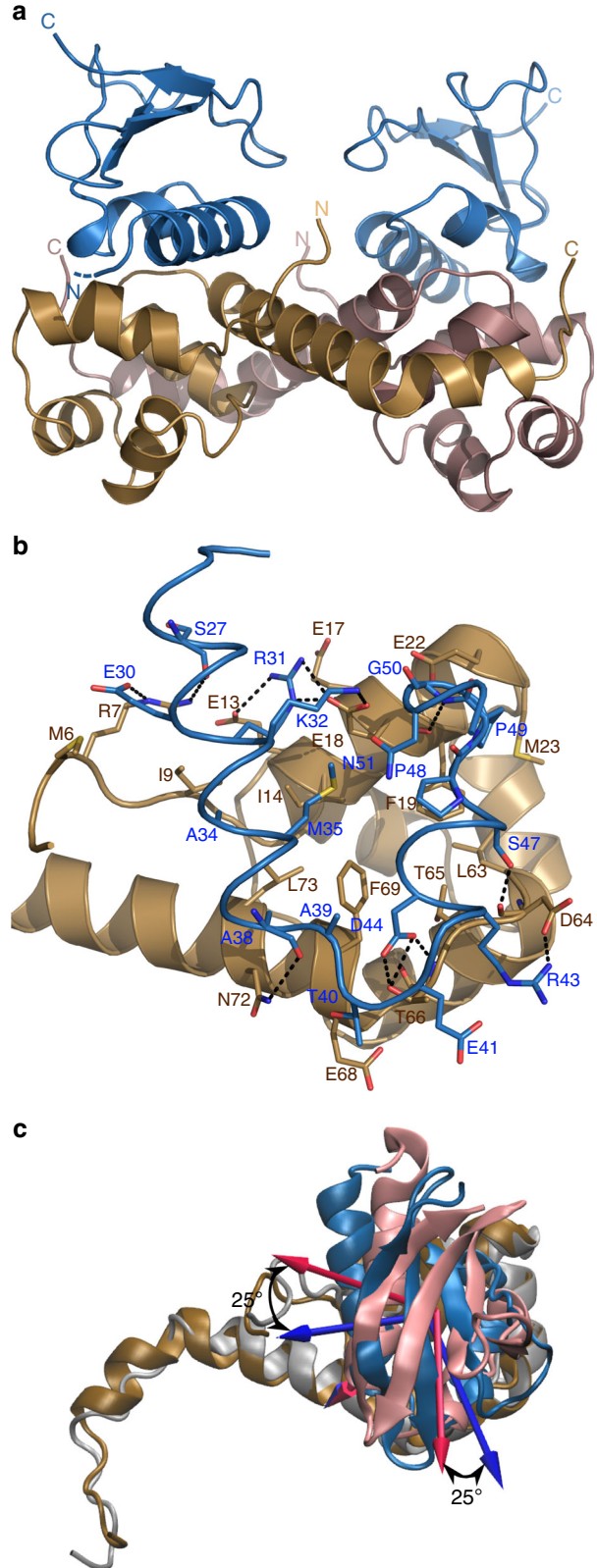

**Fig. 4** Structure of the GraTA complex. **a** Cartoon representation of the GraTA complex, which shows a GraT-GraA$_2$-GraT architecture. GraT is colored in blue and GraA colored as in Fig. 1a. N- and C-termini are labeled. The direction of the disordered extension at the N-terminus of GraT is indicated by a dashed line. **b** Zoom-in of the GraT-GraA interface. Residues which are at less than 4 Å from the other molecule are represented in stick and labeled. Colors are as in (**a**). Hydrogen bonds are shown as dashed lines. **c** Superposition of a single GraA-GraT unit from the GraTA complex on an equivalent HigA-HigB unit from *P. vulgaris* HigBA. GraA and GraT are colored as in (**a**). HigA is in gray while HigB is in pink. The 25° relative rotation of both toxins is clearly visible

the experimental scattering curve with a $\chi^2 = 1.4$ (Fig. 5b, blue curve). This four-states model shows an $R_g$ distribution with one major peak from 21.5 to 24.5 Å with weight of 0.86 (Supplementary Figure 6A). For the latter option, we modeled the 22 N-terminal amino acids in the conformation displayed by the corresponding segment of *P. vulgaris* HigB and kept the remainder of GraT as in the GraT$_2$A$_2$ crystal structure, and again compared this model to the experimental scattering curve (Fig. 5b, red curve). In this case the fitting of the theoretical to the experimental scattering curve was worse with a $\chi^2 = 10.5$. All this together suggests that despite the shape of the dimensionless Kratky plot, the N-terminal segment of GraT might indeed be disordered but is not picked up in this plot because this region only represents 10% of the protein complex.

To corroborate the previous hypothesis, we expressed and purified and inactive form of the toxin. This form consists of the substitution of the C-terminal H92 by a TEV cleavage site followed by a histidine tag (ENLYFQGSAGHHHHHH). After proteolysis with TEV this new version of GraT is only five amino acids longer than the wild type (<6% difference). In this case, the dimensionless Kratky plot (Fig. 5c) shows a slight disorder with a small shift of the maximum towards higher values (2.0; 1.22) with respect to the maximum for globular proteins with coordinates (1.73; 1.1). Furthermore, the theoretical scattering curve of the disordered ensemble fits much better the experimental scattering curve than the HigB-like conformation (Fig. 5d), with $\chi^2 = 3.1$ and 8.9, respectively. In this case, the ensemble consists on a four-state model with a major peak in the $R_g$ distribution from 13 to 15 Å with a weight of 0.8 and three other small peaks at larger $R_g$s with weights smaller than 0.1 (Supplementary Figure 6B).

**The GraT N-terminus is required for mRNase activity.** The *P. vulgaris* HigB has been shown to cleave mRNAs in a ribosome-dependent manner[29]. Thus, we tested whether GraT is also an mRNAse and whether the N-terminal region is required for its functionality. We overexpressed GraT and a truncated version of GraT without the first 22 residues (Δ22GraT) in *E. coli* and analyzed the highly abundant *lpp* mRNA for degradation. To avoid possible cross-activation of *E. coli* chromosomal TA systems in response to GraT expression, the primer extension of the *lpp* mRNA was carried out in the *E. coli* mRNase-deficient Δ10 TA strain[30]. As plasmids carrying only the *graT* gene were toxic even without induction, we modified the *graTA* operon by replacing the native Shine–Dalgarno (SD) sequence in front of the *graT* with a more effective SD from the pET11c expression plasmid. This strong SD sequence in the constructed pBBRlacI-tac-*graTA* plasmid enables expression of GraT at a higher level than GraA, and causes an inducible cold-dependent growth defect in *E. coli*. Notably, expression of Δ22GraT was not inhibitory to *E. coli* growth. GraT induction results in a distinct cleavage pattern on *lpp* mRNA, which is not detected in non-induced samples

experimental scattering data: a model with a disordered N-terminus or a model with the N-terminus folded as in *P. vulgaris* HigB. For the former option we used the online program MultiFoxs[62]. By defining the first 22 amino acids of GraT as disordered (flexible), this software generated 10,000 conformations from where it selected a four-states model as the best fit to

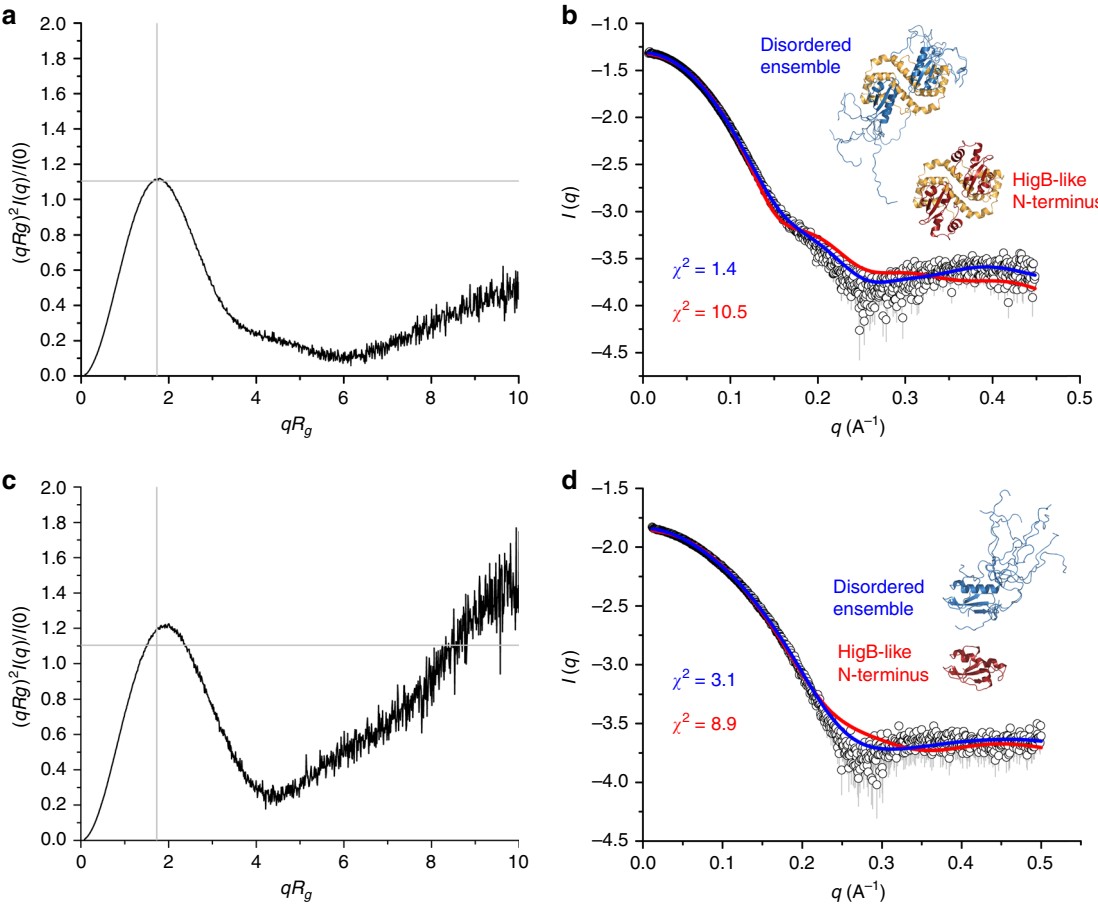

**Fig. 5** The N-terminus of GraT is intrinsically disordered in solution. Small-angle X-ray scattering (SAXS) experiment of the GraTA complex (**a**, **b**) and of the toxin GraT alone (**c**, **d**). **a**, **c** Representation of the dimensionless Kratky plot (($qR_g)^2I(q)/I(0)$) vs $qR_g$) for GraTA and GraT, respectively. The intercept of the gray lines ($\sqrt{3}$, 1.104) marks the position of the maximum for an ideal globular protein. **b**, **d** Shown are the comparison of the experimental data (open circles) with the one computed from the N-terminal HigB-like model (red) and with the profile calculated from the multi-states model obtained by the program MultiFox (blue). **b** Shown are GraT in complex with GraA and **d** GraT alone

(Fig. 6, source data are provided as a Source Data file). Most of the observed cut sites are located between the 2nd and 3rd nucleotides of codons with adenine in the 2nd position. The only exception is the 10th cleavage (Fig. 6), which is found between two codons but still following an A nucleotide. Strong codon dependence suggests that, like other HigB family toxins, GraT acts as a ribosome-dependent mRNase. Given that induction of Δ22GraT did not result in mRNA cleavage (Fig. 6), the N-terminal disordered region is necessary for GraT mRNAse activity.

In order to confirm this result in *P. putida*, we constructed the Δ22*graT* derivative strains from *P. putida* wild-type and Δ*graA* and compared the bacterial growth at different temperatures. GraT inhibits growth of *P. putida* at 20 °C in the absence of GraA, while at higher temperatures, growth is gradually restored (Supplementary Figure 7). The Δ22*graT*-encoding strains grow just like the wild-type *P. putida*, independently of the presence of the *graA* gene. Thus, GraT-mediated growth arrest in *P. putida* requires the full-length protein including its disordered N-terminus.

**GraT disorder prevents GraTA binding the operator in vitro.** GraA$_2$ strongly binds to its operator in a 2:1 molar ratio ($K_{macro}$ = 3.1 × 10$^{13}$ M$^{-2}$, Table 1, see above). By contrast, the macroscopic binding constant of the GraT$_2$A$_2$ complex for its operator as determined by ITC is significantly lower ($K_{macro}$ < 10$^7$ M$^{-2}$)

(Fig. 7a). This was not expected because superimposition of the GraT$_2$A$_2$ complex onto the GraA$_2$-DNA-GraA$_2$ complex shows that GraT binds to GraA at an opposite side relative to the DNA binding site of GraA (Fig. 1b), and that there is no direct steric interference between GraT and the DNA (Supplementary Figure 8).

Two structural properties of the GraT$_2$A$_2$ complex may contribute to the low affinity of GraT$_2$A$_2$ for the operator: (1) the 2.2° re-orientation of the two GraA monomers when bound to GraT (together with a possible function-interfering alteration in the dynamics of GraA$_2$ upon GraT binding), and (2) the presence of the intrinsically disordered segment at the N-terminus of GraT. To distinguish between these possibilities, we tested the binding of Δ22GraT$_2$A$_2$ (where GraT is lacking the first 22 residues) to the *graTA* operator using ITC (Fig. 7b). Δ22GraT$_2$A$_2$ binds to the operator even tighter ($K_{macro}$ = 2.3 × 10$^{15}$ M$^{-2}$) than GraA$_2$ alone ($K_{macro}$ = 3.1 × 10$^{13}$ M$^{-2}$, Table 1), indicating that GraT does not lock GraA$_2$ into a binding-incompetent conformation. EMSAs using different (GraT$_2$A$_2$ or Δ22GraT$_2$A$_2$):operator molar ratios further confirm that in contrast to the wild-type GraT$_2$A$_2$, the Δ22GraT$_2$A$_2$ complex readily binds the *graTA* operator (Fig. 7c, d). These experiments suggest that the intrinsically disordered N-terminus of GraT is directly responsible for impairment of operator binding, likely via steric hindrance.

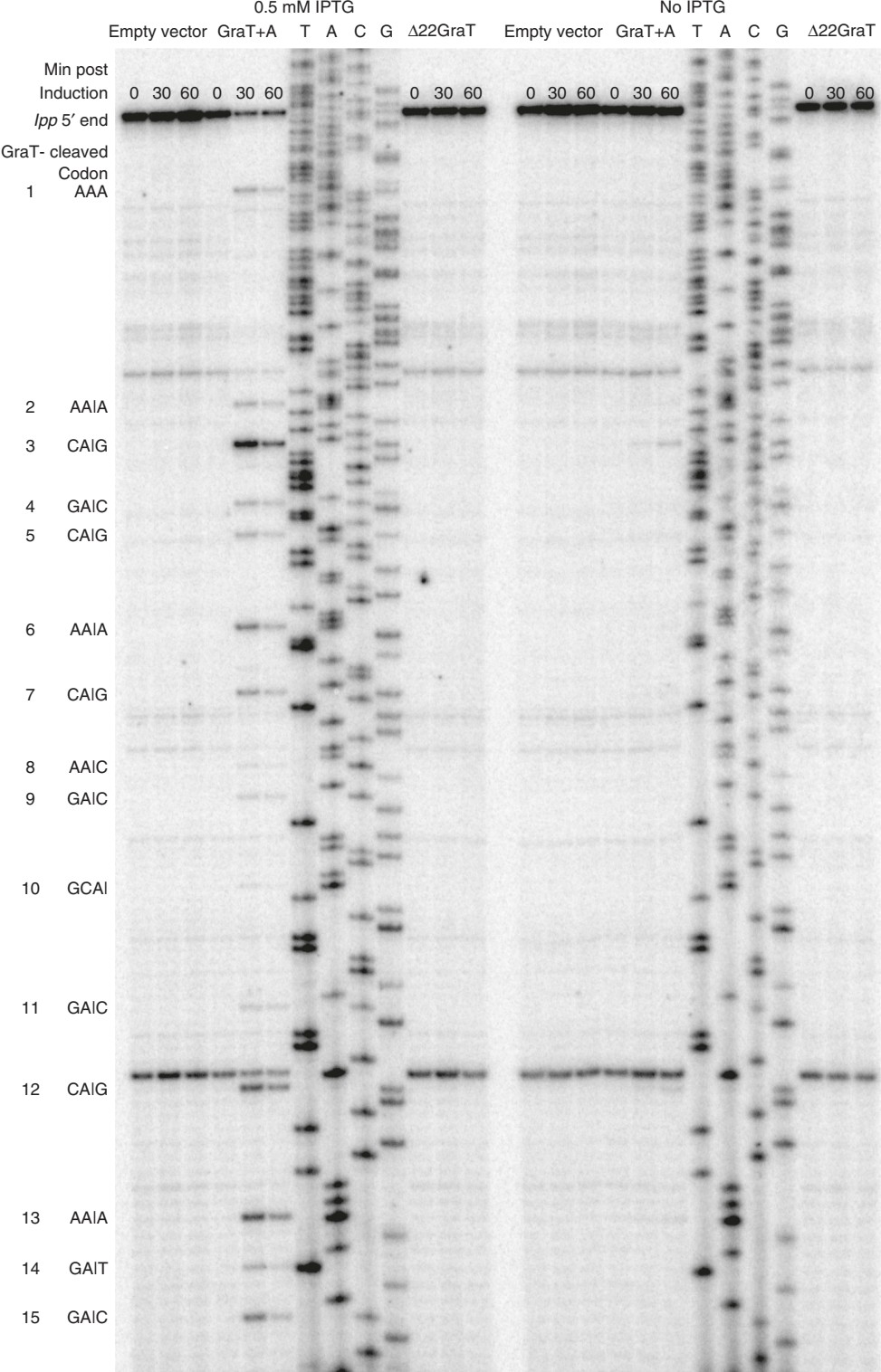

**Fig. 6** The disordered N-terminus of GraT is required to cleave mRNA. Autoradiography of the *E. coli lpp* mRNA primer extension analysis. MG1655 Δ10TA cultures carrying either the empty vector pBBRlacItac, pBBRlacItac-graTA or pBBRlacItac-Δ22graT were sampled 0, 30 and 60 min after GraT induction at 20 °C (non-induced samples are included as controls). Inducing the full GraT but not the Δ22 mutant variant causes mRNA cleavage. The cut sites follow adenines, usually in the 2nd position of codons. Cleaved codons are shown next to the gel image with cleavage sites, if precisely determined, represented by vertical lines

**GraTA overexpression derepresses the *graTA* promoter in vivo.** The role of GraT in derepression of the *graTA* promoter was further tested in vivo. The *graT-lacZ* transcriptional fusion was used as a reporter for *graTA* promoter activity in *E. coli*. The genes coding for GraTA, GraA or Δ22GraTA were cloned under

an isopropyl β-D-1-thiogalactopyranoside (IPTG)-inducible *tac* promoter. For enabling expression of GraT at a higher level than GraA, the native SD sequence in front of *graT* was replaced with strong SD from the pET11c expression plasmid. Note that for expression of the GraTA complex, the same plasmid was used as

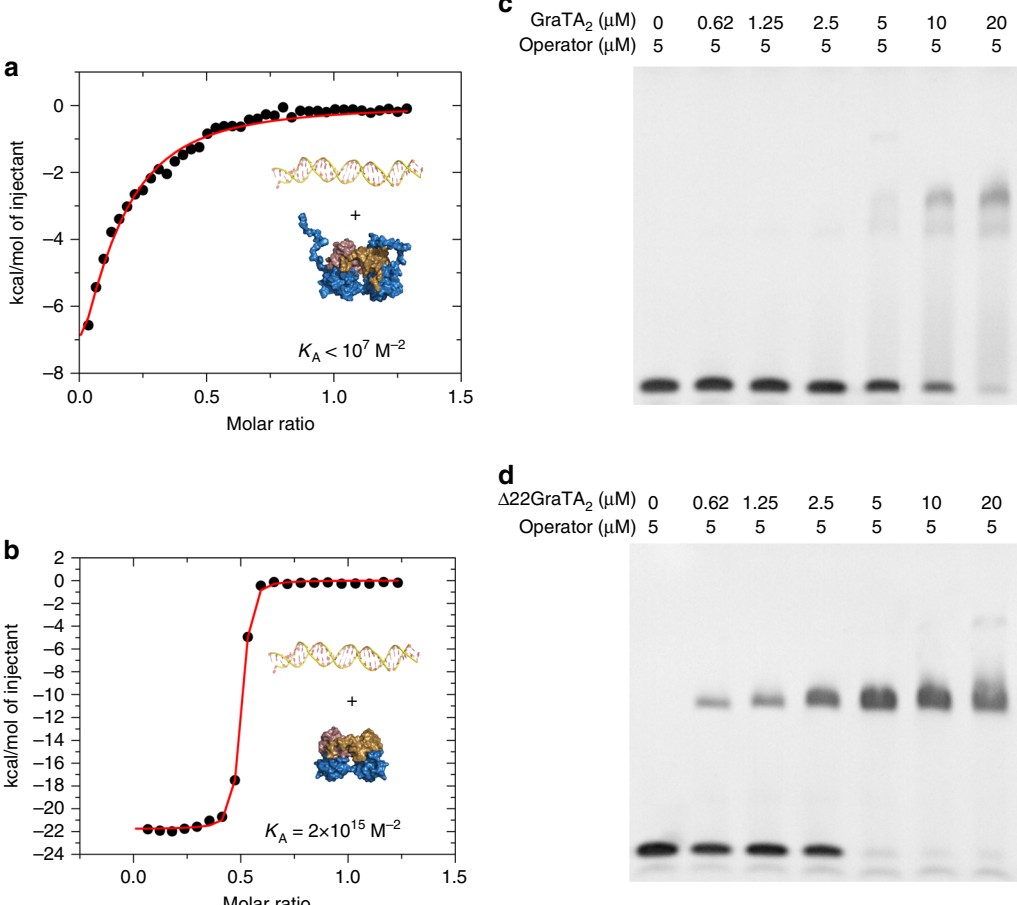

**Fig. 7** The flexible N-terminus of GraT inhibits $GraT_2A_2$ binding to the operator in vitro. **a** Isothermal titration calorimetry (ITC) titration of *graTA* operator DNA into wild-type $GraT_2A_2$ complex. **b** ITC titration of *graTA* operator into the truncated $\Delta22GraT_2A_2$ complex. **c** Electrophoretic mobility shift assays (EMSAs) using different $GraT_2A_2$/operator ratios (0:1, 1:8, 1:4, 1:2, 1:1, 2:1 and 4:1). **d** EMSA with $\Delta22GraT_2A_2$ and *graTA* operator using the same protein/DNA ratios as in (**c**)

in the mRNase assay. The plasmids were independently transformed into *E. coli* already containing the *graT-lacZ* reporter. β-Galactosidase measurements show that while GraA alone represses the *graTA* promoter activity, the expression of the GraTA complex leads to full derepression of the promoter (Fig. 8a, source data are provided as a Source Data file). Expression of the Δ22GraTA with N-terminally truncated GraT again represses the promoter (Fig. 8a), which is in good accordance with strong binding of $\Delta22GraT_2A_2$ to the operator (Fig. 7). Expression of the antitoxin and toxin proteins was verified by western blot (Fig. 8b). These results confirm the in vitro data that the GraT N-terminal segment is hindering the binding of GraTA complex to the operator and that GraT acts as a derepressor of the *graTA* operon transcription.

## Discussion

In Eukaryotes, many proteins contain long stretches of functional unstructured regions that vary in size from dozens to hundreds of amino acids. This lack of structure allows for activities that are difficult to perform by well-folded proteins. Typical intrinsic disorder-specific functions involve hub proteins moonlighting different tasks[31,32], the separation of affinity and specificity in folding-upon-binding events[33,34], molecular clocks[35,36] and springs[37], entropic barriers and counting of phosphate groups[38]. Intrinsic disorder is common within transcription regulators in Eukaryotes[39,40]. In transcription factors from Prokaryotes, intrinsic

disorder is abundantly present in antitoxins from toxin–antitoxin modules[7,21]. The majority of structurally characterized antitoxins contain a disordered region that neutralizes its toxin counterpart by folding upon binding. This region is also crucial to allow rapid degradation of the antitoxin upon activation of Lon or ClpXP proteases as initially observed for F-plasmid CcdA[41,42] and later for many other antitoxins[7,21,22]. In the case of the well-studied *phd/doc* module from bacteriophage P1, the intrinsically disordered segment at the C-terminus of Phd prevents the binding of two Phd dimers adjacent to each other on the operator in the absence of Doc via a mechanism of entropic exclusion[43]. This feature helps to enhance the effect of conditional cooperativity[10,11], the mechanism that tightly regulates the TA operon transcriptional activation preventing random expression of toxin[44–46].

Different from many other TA loci, the *graTA* operon is not regulated by conditional cooperativity. GraA alone is responsible and sufficient for operon autorepression, while the toxin GraT leads to transcriptional derepression when it binds the antitoxin. This behavior of GraT resembles that of *E. coli* toxin MqsR which also destabilizes the antitoxin-DNA complex[17], yet the molecular mechanisms of two toxins differ. For MqsA, binding to DNA or to the toxin is mutually exclusive because the DNA and toxin binding sites overlap on the antitoxin[17]. In contrast, the binding sites of GraA for toxin and DNA are located at opposite surfaces of the protein.

The GraTA module is also unique in that the toxin contains an intrinsically disordered segment that is involved in regulation of

**a**

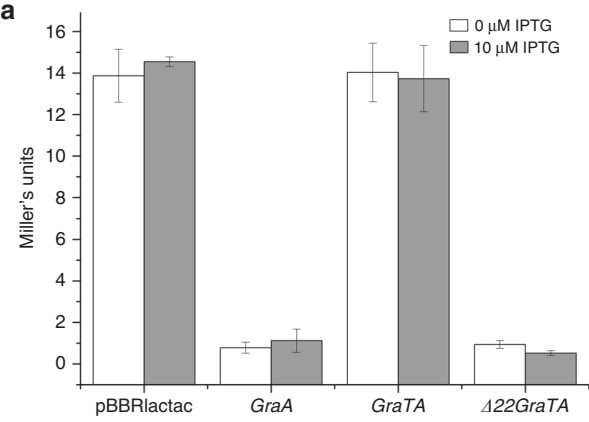

**b**

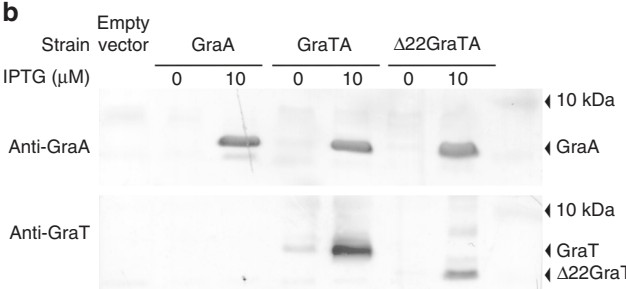

**Fig. 8** The flexible N-terminus of GraT is required for derepression of the *graTA* promoter. **a** β-Galactosidase activities measured in four *E. coli* strains. Each strain contains the *graT-lacZ* transcriptional fusion in plasmid p9TT1586 and one of the following plasmids: an empty vector (pBBRlactac), pBBRlactac-*graA*, pBBRlactac-*graTA* or pBBRlactac-Δ22graTA. Values without induction (white) and after induction of GraA, GraTA or Δ22GraTA proteins with 10 μM of isopropyl β-D-1-thiogalactopyranoside (IPTG; gray) are presented. Error bars indicate 95% confidence intervals of at least three independent experiments. **b** Western blots detecting GraA (upper panel) and GraT (lower panel) in the *E. coli* strains that were tested in the β-galactosidase assay

transcription. The 22 N-terminal residues of GraT are disordered in both the isolated state and the GraTA complex and prevent the TA complex from interacting with the operator. Given that the N-terminally truncated version of GraT does not prevent the interaction of GraA with its operator, the 22-amino-acid intrinsically disordered sequence at the N-terminus of GraT seems to be a major contributor to the inability of the GraT-GraA complex to bind its target DNA sequence. The charge distribution of this segment does not seem to be responsible for the GraT-DNA repulsion. Out of these 22 residues, only 6 are charged, 4 positively and 2 negatively, yielding a net charge of +2, which would actually attract the negatively charged DNA. On the other hand, this disordered stretch is more likely to work as an entropic barrier that sterically impedes the formation of the $GraT_2A_2$-DNA complex. The N-terminus of the globular part of GraT in a virtual GraTA-DNA complex points towards the DNA (Supplementary Figure 8), and binding of the GraTA complex to the operator would heavily restrict the conformational ensemble that can be adopted by the disordered GraT N-terminus.

Interestingly, while the overall structures of GraA and GraT are highly similar to the HigA and HigB proteins of *P. vulgaris*[16], the N-terminal segment in the HigB toxin is, differently from GraT, fully structured. This ordered HigB N-terminus does not prevent interaction of HigBA with the operator as it folds back in the opposite direction, contacting the N-terminus of HigA[16]. This

structural divergence between the two toxins results in different outcome in transcription regulation, as HigB does not inhibit HigA binding to the *higBA* operator[16]. This vividly demonstrates the importance that intrinsically disordered regions play in protein functionality.

Binding of GraA to its operator is highly cooperative. Interestingly, this cooperativity functions via a communication pathway involving only the DNA partner and without direct contacts between the two bound GraA dimers. It has been described that the mechanical properties of DNA indeed allow for transmission of allosteric effects via local distortions of the major groove (and to a lesser extent the minor groove)[47]. The DNA distortions that affect the binding dissociation kinetics are dampened as a function of the distance between the proteins bound to the DNA[47]. QacR and PA2196 are two transcriptional regulators of the TetR family, for which such effects are well documented and that show the same arrangement as $GraA_2$ when bound to DNA[48,49] (Supplementary Figure 9). Their interaction with DNA also involves cooperativity between two non-interacting dimers, and this cooperativity is achieved via a 4 Å stretching of the DNA[48,49]. However, $GraA_2$ does not induce any significant deformation in the operator region within experimental errors. Allostery is thus likely mediated by an accumulation of relatively small DNA deformations and/or changes in DNA dynamics.

The N-terminal 22-amino-acid segment of GraT toxin not only affects operator binding of the GraTA complex, but also its own activity. The latter is unexpected given that the predicted active site near His92 is very similar to the active site of HigB and the N-terminus does not likely contribute catalytically important residues. While the 22 N-terminal residues are flexible and therefore might be considered to move towards the active site, this would likely prevent productive binding to the ribosome.

Therefore, it is more likely that upon ribosome binding this region would fold in such a way that it facilitates the interaction between the ribosome and GraT without directly being involved in catalysis. In line with that, the crystal structure of *P. vulgaris* HigB bound to the ribosomal 30S subunit demonstrated that the first N-terminal α-helix of HigB, the very same region that is unstructured in GraT, is important for HigB recognition of the 16S rRNA in the ribosomal A site[28]. Given that several substitutions in HigB α1 decrease HigB toxicity by disturbing its ribosome binding[28], it is reasonable to assume that the N-terminus of GraT also plays a role in ribosome binding.

In general, disordered regions in proteins may be prone to degradation. Fast degradation of antitoxins after activation of Lon during episodes of stress is thought to be facilitated by the presence of intrinsic disorder[7,21] and is crucial for the regulation of many toxin–antitoxin modules[22]. In contrast with most antitoxins, GraA is a fully ordered protein and, in accordance with that, is also uncommonly stable and not degraded by Lon or Clp[20]. Still, the lifetime of GraA is sensitive to the bacterial growth phase and ATP levels[20], indicating that GraA degradation can be triggered by certain conditions. Interestingly, MqsA, a fully structured antitoxin like GraA, is also highly stable under normal growth conditions but is quickly degraded under oxidative stress[50]. Although speculative, the finding that the toxin GraT rather than the antitoxin GraA contains the intrinsically disordered region raises an intriguing possibility of the toxin GraT being under proteolytic control. Preferential degradation of the toxin would not only lead to depletion of GraT toxicity but also to the presence of free antitoxin dimers that prevent transcription. This would indeed be a strong mechanism that protects *Pseudomonas putida* against accidental *graTA* activation and might contribute to the mild and temperature-dependent effects of GraT toxin[18]. However, as the stability properties of the GraT and the trigger for GraA degradation are currently unknown, the

conditions under which the GraTA module becomes active and the molecular mechanism behind such activation remain to be determined.

## Methods

**Construction of plasmids and strains.** Construction of the plasmids and strains used is detailed in the Supplementary Methods. The different plasmids, strains and oligonucleotides are listed in Supplementary Table 2.

**Protein expression and purification.** HisGraA and HisGraTA were expressed in *E. coli* harboring recombinant pET11c-derived plasmids[51]. In the case of GraA alone, it was N-terminally His-tagged, while for the complex GraT it was His-tagged at the N-terminus and GraA was left untagged. Similar constructs were generated for Δ22GraTA-his and hisTEV-GraTA by amplifying the modified *graTA* operon with either the GraT_22_Nde/A-his or hisTEV-graT/1585Bam oligonucleotide pair, respectively, and cloning into the *Nde*I/*Bam*HI-opened pET11c. The gene *grat_tev_6xhis* was chemically synthesized and cloned into *Nde*I/*Not*I-cleaved pET-21b (Supplementary Table 2). In all cases, cells were first grown at 37 °C until optical density (OD)~0.6 when the temperature was lowered to 20 °C and protein expression was induced with 0.5 mM IPTG for 16 h.

Cells were lysed in 50 mM Tris, pH 8.0, 250 mM NaCl and 2 mM β-mercaptoethanol (Buffer A) and protease inhibitors. Purifications were carried out by loading each cell lysate supernatant onto a Ni-Sepharose column. The proteins were eluted with Buffer A plus 500 mM imidazol using a step gradient. The protein containing fractions were subsequently loaded into a Bio-Rad SEC 70 gel filtration column equilibrated with 50 mM Tris, pH 8.0, 250 mM NaCl and 2 mM β-Mercaptoethanol. When purifying the proteins for SAXS or ITC, the 2 mM β-mercaptoethanol was substituted by 2 mM tris(2-carboxyethyl) phosphine (TCEP).

**Preparation of double-stranded DNA.** The *graTA* operator region and its variants were obtained from single-stranded oligonucleotides purchased from Sigma-Aldrich. The oligonucleotides were resuspended in water at ~200 μM. Complementary strands were mixed in a 1:1 ratio to obtain a final concentration of ~100 μM of the corresponding double-stranded DNA. The mixture was heated to 80 °C for 20 min and slowly cooled to room temperature. The formation of all double-stranded DNA was checked by gel filtration.

**Crystallization and structure determination.** HisGraA$_2$, hisGraT$_2$A$_2$ and the hisGraA$_2$/*graTA* operator complex crystallized in different conditions[51] (Supplementary Table 3). All datasets were collected at 100 K at Proxima 1 beamline (Soleil Synchrotron, Gif-Sur-Yvette, France) at 1.9, 2.2 and 3.8 Å resolution for hisGraA$_2$, hisGraT$_2$A$_2$ and hisGraA$_2$/*graTA* operator complex, respectively.

All datasets were processed with XDS[52] using the XDSME interface. All the structures were determined by molecular replacement with the program PHASER[53] from the CCP4 program suite[54]. For GraA$_2$, the HigA protein from *Coxiella burnetii*[23] (PDB code: 3TRB) was used as search model. For the structure of GraT$_2$A$_2$, the previously solved GraA$_2$ structure together with HigB from *Proteus vulgaris*[16] (PDB code: 4MCX) were used as search models. The refined GraA$_2$ structure and the coordinates of an in silico generated structure of the *graTA* operator (AAATTAACGAATAACGTTAAGCATTCAGCTCAT) were used as search models in the structure determination of the (GraA)$_2$-*graTA* operator complex. Several rounds of refinement and model building were performed with phenix.refine[55] from the PHENIX package[56] and COOT[57]. The quality of the structures was checked with MolProbity[58]. Data collection and refinement statistics are shown in Supplementary Table 1.

**Small-angle X-ray scattering.** SAXS data were collected at SWING beamline (Soleil Synchrotron, Gif-Sur-Yvette, France). This beamline has a SEC system before the measuring capillary. SEC will remove possible aggregates rendering a very homogeneous sample that will then be directly exposed to X-rays for data collection. All the experiments were performed in 50 mM Tris, pH 8.0, 250 mM NaCl and 2 mM TCEP as running buffer.

Each protein was run through a Shodex KW402.5-4F at 0.2 ml/min. Scattering curves covering a concentration range around the peak were normalized and averaged to obtain the final scattering curve. $R_g$ values were derived from the value of $I_0$ which were obtained by extrapolating to $q = 0$ using the Guinier approximation as implemented in ATSAS suite[59]. The molecular weights of the different entities were estimated in a concentration independent way using the $I_0$, Porod volume and Fisher methods.

The use of the dimensionless Kratky plot (($qR_g$)$^2I(q)/I(0)$ vs $qR_g$) is a relatively easy way to show that a protein is completely folded, partially folded or completely unstructured[60]. If a protein is globular it follows Guinier's law $I(q)/I(0) = \exp(-(qR_g)^2/3)$. The corresponding dimensionless Kratky plot is a function $f(x) = x^2\exp(-x^2/3)$, with $x = qR_g > 0$ with maximum of 1.104 at $qR_g = \sqrt{3}$. On the other hand, an ideally disordered protein follows Debye's law $I(q)/I(0) = 2(x^2-1-\exp(-x^2))/x^4$, with $x = qR_g > 0$. In this case the Kratky plot is described by the function $f(x) = 2(x^2-1-\exp(-x^2))/x^2$ which increases monotonically with an asymptote at $f$

$(x) = 2$. Experimentally, globular proteins show a very similar normalized Kratky plot with a maximum at ($\sqrt{3}$; 1.1), while partially unstructured proteins show a maximum shifted to higher values in both axes[61].

The first 22 amino acids of GraT that are missing in the GraT$_2$A$_2$ structure were computationally modeled using the program Modeller[63]. We used the webserver MultiFoxs[62] (http://modbase.compbio.ucsf.edu/multifoxs/) to generate 10,000 conformations of GraTA by sampling the conformational space of the first 22 residues while keeping the rest of the complex rigid. Then, theoretical SAXS profile is calculated for each conformation to further perform a multi-state model enumeration.

All SAXS data and the corresponding analysis are summarized in Supplementary Tables 4 to 8.

**Isothermal titration calorimetry.** All ITC titrations were carried out in an iTC200 calorimeter (GE Healthcare). Prior to the measurements, GraA$_2$, GraT$_2$A$_2$, Δ22GraT$_2$A$_2$ and the various dsDNA fragments were dialyzed against 50 mM Tris pH 8.0, 250 mM NaCl and 2 mM β-Mercaptoethanol to minimize buffer mismatch. For GraA$_2$ and Δ22GraT$_2$A$_2$ the concentrations in the 200 μL cell were 20 and 18 μM, respectively. Several injections of 2 μL of *graTA* operator at 90 μM were used. The concentration of GraT$_2$A$_2$ was 165 μM and *graTA* operator was injected in 1 μL volumes at 989 μM. All the titrations were performed at 25 °C. Raw data were integrated and corrected for the buffer dilution heat effects using the MicroCal Origin software to obtain the enthalpy change per mole of added ligand corrected for the buffer dilution effects. Calorimetric isotherms were analyzed using a model describing a cooperative binding of the protein species to the *graTA* operator:

GraA2+1/2 operator → 1/2 GraA$_2$-operator.

Model-adjusted parameters Δ$G_A$ (free energy of association per mole of ligand) and Δ$H_A$ (standard enthalpies of association per mole of ligand) were obtained through least-square fitting as implemented in MicroCal Origin 7.0 and SEDPHAT[64]. Equilibrium association constants $K_A$ reported in the text were calculated from the corresponding Δ$G_A$.

**Electrophoretic mobility shift assay.** Assays were performed by mixing different protein concentrations (0, 0.625, 1.25, 2.5, 5, 10 and 20 μM) and a fixed *graTA* operator concentration (5 μM) to obtain the following protein/DNA molar ratios: 0:1, 1:8, 1:4, 1:2, 1:1, 2:1 and 4:1. The mixtures were incubated at room temperature for 30 min and loaded onto a native 6% polyacrylamide gel prepared with TBE (Tris/Borate/EDTA) buffer. The gel was run embedded in ice with pre-cooled TBE as running buffer (the first 10 min at 180 V followed by 25 min at 120 V). The DNA was stained with ethidium bromide and visualized using a Proxima 10 Phi (ISOGEN Lifescience) apparatus.

**TEV protease digestion.** A TEV cleavage site (GNLYPQ|G) was inserted between the 6xHis tag and the second amino acid of GraT. His-TEV-GraT$_2$A$_2$ was incubated with hisTEV overnight at a molar ratio of 5:1. The reaction product was incubated for 10 min with Ni-sepharose resin and centrifuged at 5000 rpm. hisTEV, the 6xHis containing peptides and uncleaved protein bound to the resin, while tag-less GraT$_2$A$_2$ was collected from the supernatant. Completion of the reaction was confirmed by sodium dodecyl sulfate–polyacrylamide gel electrophoresis (SDS-PAGE) and western blot using a anti-Histidinge-tag antibody.

**Native mass spectrometry.** Samples of GraA$_2$-DNA for native ion mobility mass spectrometry[65] were prepared by mixing together GraA$_2$ dimer and DNA duplex in 40 mM aqueous ammonium acetate pH 7 to provide complexes with 1:1, 2:1 and 4:1 stoichiometry, and this each time with a final concentration of GraA$_2$ dimer of 20 μM. The samples were introduced into the mass spectrometer using nano-electrospray ionization using in-house prepared gold-coated glass capillaries and a spray voltage of +1.6 kV. Spectra were recorded on a traveling wave Q-TOF instrument (Synapt G2, Waters, Manchester, UK) tuned for transmission of large, native protein assemblies. Voltages used were sampling cone 50 V, trap collision cell 75 V and trap DC bias 45 V. Pressures in subsequent stages of the instrument were 5 mbar, $2.86 \times 10^{-2}$ mbar and 3.45 mbar for source, trap cell and ion mobility cell, respectively. Spectra were externally calibrated using a 10 mg/mL solution of cesium iodide. Analyses of the acquired spectra were performed using Masslynx version 4.1 (Waters, Manchester, UK).

**mRNA degradation assay.** *E. coli* MG1655 Δ10 TA was transformed with either pBBRlacItac, pBBRlacItac-graTA or pBBRlacItac-Δ22graT. Overnight cultures were diluted into 50 mL fresh LB medium to OD$_{580}$ of ~0.1 and grown at 30 °C. After 1 h, temperature was shifted to 20 °C. After 30 min, the cultures were split in two parts and 0.5 mM IPTG was added to one parallel. Then, 1.5 mL of cultures were harvested right before IPTG addition at 30 min and 60 min post induction. Cells were pelleted and frozen in liquid N$_2$. RNA was extracted using RNAzol® RT (Molecular Research Center, Inc.). For primer extension, the oligonucleotide lpp_ec was radiolabelled using [γ-$^{32}$P]ATP (Hartmann Analytic) and the reactions were carried out using AMV (avian myeloblastosis virus) reverse transcriptase (Promega). The reference sequence was amplified from the *E. coli* genome with lpp_ec and lpp_ec_ees and sequenced with radiolabelled lpp_ec using the

Sequenase™ Version 2.0 DNA Sequencing Kit (Affymetrix). The resulting fragments were separated on a 7% polyacrylamide-urea gel and visualized on a Typhoon phosphoimager (GE Healthcare).

**Temperature-sensitive growth assay.** For assaying the in vivo functionality of GraT and its N-terminal truncation derivative Δ22GraT, the *P. putida* wild-type PaW85, Δ*graA* and their Δ22*graT* derivative strains were grown overnight in LB medium. Overnight cultures were diluted 100-fold, spotted onto LB plates as 5 μL drops and incubated at 20 °C, 25 °C and 30 °C for 24 h.

**β-Galactosidase assay.** *E. coli* DH5α cells harboring *graT-lacZ* transcriptional fusion in plasmid p9TT1586 were transformed with one of the plasmids for expression of GraTA proteins (plasmids pBBRlacItac-graA, pBBRlacItac-graTA and pBBRlacItac-Δ22graTA) or with an empty plasmid pBBRlacItac. At least three individual fresh transformant colonies were inoculated into LB media containing gentamycin (final concentration 10 μg/mL) and ampicillin (100 μg/mL). For induction of GraTA proteins, 10 μM IPTG was added and bacteria were grown for 20 h at 37 °C. β-Galactosidase activities were measured by a protocol described previously[18]. Expression of GraA and GraT proteins was verified by western blot analysis. For that, proteins were separated on 10% Tricine-SDS-PAGE and transferred to polyvinylidene difluoride membranes. GraA and GraT were detected by probing membranes with anti-GraA and anti-GraT polyclonal antibodies, respectively. The blots were treated with alkaline phosphatase-conjugated anti-mouse immunoglobulin G and developed using bromochloroindolyl phosphate/nitro blue tetrazolium.

## Data availability

All the crystallographic structural data has been deposited into the Protein Data Bank (www.rcsb.org) with the following accession codes: 6F8H (GraA$_2$), 6F1X (GraA$_2$/DNA) and 6F8S (GraA$_2$T$_2$). The scattering data produced by the SAXS were deposited at the www.sasbdb.org database with the accession codes: SASDE68 (GraA-Operator complex), SASDE58 (GraTA complex), SASDE48 (GraT). The raw data files supporting Figs. 6 and 8a are included in the source data file. Other data are available from the corresponding authors upon reasonable request.

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

## Acknowledgements

This work was supported by grants from the Fonds voor Wetenschappelijk Onderzoek Vlaanderen (grant numbers G.0135.15N, G0C1213N, G.0090.11N); the Onderzoeksraad of the Vrije Universiteit Brussel (grant numbers OZR2232 to S.H., SPR13); the European Community's Seventh Framework Programme (FP7/2007-2013) under BioStruct-X (projects 1673 and 6131); the Hercules Foundation (grant number UABR/11/012); the Estonian Research Council (grants IUT20-19 and PUT1351) and the Slovenian Research Agency (Grants P1–0201 and J1-5448). Funding for open access charge: Estonian Research Council grant PUT1351.

## Author contributions

A.T. and R.L. designed the study. A.T., R.L. and R.H. wrote the manuscript and the answers to the referees. A.T. performed the crystallography studies, ITC, SAXS and EMSA experiments together with the protein expression and purification. H.T. and R.H. constructed most of the plasmids, the study about the derepression of the *graTA* promoter and the in vivo toxicity of GraT. A.A. performed the studies on the mRNase activity of GraT. A.K. and F.S. performed the native mass spectrometry studies. A.G.-P. largely contributed to the interpretation of the data. S.H. contributed with the processing of the ITC data.
