## [Peer Review File · Nature Communications]

Reviewers' comments:

Reviewer #1 (Remarks to the Author):

The manuscript submitted by Talavera et al. reports a structural and functional studies on the GraA GraT antitoxin toxin system. The authors describe their experimental evidence for GraA binding to the promoter DNA from which they evolve a model for cooperative binding of two GraA-dimers on opposing sides of the palindromic sequence within the GraA promoter region. Interestingly, no direct specific readout of the DNA sequence is established as not a single contact between residues of GraA and any base moiety is observed in the Gra2/DNA crystal structure. Thus, the authors speculate about a kind of allosteric signaling mediated by the DNA resulting in the observed cooperative binding. Furthermore, the authors convincingly show that a single HTH-motif of each GraA dimer binds to the central palindromic sequence and confers specificity. In contrast, the second HTH-motifs seems to contribute only to affinity but not to specificity for the DNA. From their structural and functional studies, the authors infer that the N-terminal 21 residues of the GraT toxin impair with promoter binding. Furthermore, the authors show that removal of this region results in a non-functional toxin.

The manuscript is well written in most parts and comprehensible. However, as discussed in detail below, there are remaining unclear but important issues.

Comment 1: Title and abstract: As outlined below in detail in comment 5, it remains to be shown whether the N-terminus of GraT is indeed intrinsically disordered or whether this region is just unstructured in the GraAT complex. Furthermore, most parts of the manuscript describe the structural and functional studies on GraA promoter binding. Thus, the title might be misleading and should be revised.

Comment 2: Nature of the DNA helix in the crystal contacts. It remains unclear from the manuscript, whether the 33 bp of dsDNA forms an undisturbed entity with the crystal packing or something like a continuous helix as observed in several other protein-DNA co-crystal structures is formed. Can the authors exclude crystal-packing artifacts to cause the lack of structural features (be it bending / straightening or different rigidity) that would allow to explain sequence specificity?

Comment 3: Table 1 and Figure S3. Since the numbers are quite high, it is likely that those are presumably not Kd but KA values. I guess the latter is correct as the authors stated in the Materials and Methods section that KA values were calculated. For all ITC experiments the quality of the fit cannot be judged, and no errors are given. Additionally, has the heat of dilution been considered in ITC experiments. This is important to some extent, as the fitted entropic and enthalpic contributions are given as well.

Comment 4: Architecture of the GraAT protein complex. It would be interesting if the authors could give some information how GraA inhibits GraT activity.

Comment 5: The authors unambiguously show that the N-terminus of GraT is disordered in the GraAT complex. However, it remains to be shown whether this part of the toxin is ordered in GraT in the absence of GraA. Most importantly, it remains unclear, whether truncation of the latter in GraT makes the protein in-stable. When judging the protein levels shown in Figure 8 panel C, much less protein is detected in E. coli when compared to the full-length protein (excluding technical issues such as for instance different affinity of the polyclonal GraT-antibody for the two variants or that antibody binding regions have been removed). Is this variant less stable or less expressed in this experiment? Moreover, the authors speculate that the N-terminal region of GraT is too distal to the catalytic site to have any direct impact on activity. It would be interesting to know, what the distance between the active site and the N-terminus is. In fact, the extended model for the N-terminus shown in Figure 5 panel B would suggest that this part could wrap

around the entire surface of the molecule and reach any point at the molecular surface of GraT. Finally, could thermostability relate to the observed cold-sensitive phenotype in *P. putida* and has the truncation any effect on that?

Comment 6: Along the same line as outlined in comment 5: Given the homology with for instance HigB, it is convincing that the loss of ribosome binding could be the reason for the lack of activity when N-terminal truncated variant of GraT were used in lpp mRNA degradation experiments. Yet, can the authors provide any additional evidence for their hypothesis?

Comment 7: De-repression of the operon by GraT. Apparently, the protein levels of full-length and truncated GraT are very different in *E. coli* (excluding issues with the antibody used for the WB shown in Figure 8 panel C). Thus, the observed transcriptional repression of the operon upon expression of the N-terminal truncated GraT in the beta galactosidase assay could be mainly caused by GraA alone which is in huge excess over the truncated GraAT complex. Can the author exclude this?

Comment 8: Discussion: ...This region of GraT remains disordered when bound to the antitoxin... As the authors do not have any experimental evidence for that, it is advisable toning this statement down.

Comment 9: Similarly, ...the 22-amino acid intrinsically disordered sequence at the N-terminus seems to be solely responsible for the inability of the GraT-GraA complex to bind its target DNA sequence... and the entire residual paragraph do require some reconsideration unless further evidence supporting this hypothesis is provided.

Comment 10: ...The DNA distortions that affect the binding k_{off} are dampened ... should rather read like ... The DNA distortions that affect the dissociation kinetics are dampened ... or something like that.

Comment 11: The entire paragraph starting with ... Intrinsic disorder in the GraT toxin not only affect operator binding ... might require some reconsideration based on the lack of information about the structure GraT alone or when bound to the ribosome.

Comment 12: It was somewhat unclear from the main text that translation of GraT is initiated from an individual SD sequence. It might be worthwhile to briefly mention that fact somewhere in the manuscript.

Comment 13: Figure 1 panel B: What were the limits for the electrostatic surface potential?

Comment 14: Figure 3: What were the experimental conditions for this experiment? The authors might want to mention briefly which column they have used for size exclusion chromatography.

Comment 15: Figure 7 panel A and B: Are the given constants association or dissociation constants?

Comment 16: Table S1: The formatting of the space group symbols is not correct and ambiguous (P32 1 1 or P3 2 1. Furthermore, the PDB codes given in table do not match with the file names of the deposited validation reports.

Comment 17: As the criteria for chosen resolution cutoff of the low-resolution structure of the GraA2/DNA complex is very different when compared to the two other structures, it would be worth to show a representative electron density map demonstrating the quality of the map.

Reviewer #2 (Remarks to the Author):

A dual role in regulation and toxicity for the intrinsically disordered N-terminus of the toxin GraT

Talavera et al. have thoroughly characterized a bacterial toxin-antitoxin system with a wide range of biochemical and biophysical methods. The manuscript is clearly written and the results as well as the conclusions are well presented. The findings are interesting also for a wider audience.

Some comments regarding the structural properties of the GraT N-terminus:

* Length of 22 residues for the intrinsically disordered N-terminus;

- Are the other intrinsically disordered regions (observed for other anti-toxins) of similar length?

- Perhaps, here it could be mentioned that 22 of 92(?) residues makes up almost 1/4 of the toxin

* SAXS models as shown in Figure 5B:

- in the orientation of the upper structure it looks as if the extensions of the blue models are at the C-terminus (extension of the helix) as opposed to the N-terminus (compare to the orientation in Figure 4)

- selected ensemble: the ensemble shown here overlap pretty much. Thus, the flexibility seems rather constrained. Does the program MultiFOXS address this? Eg, the distribution of $D_{max}/R_g/Volume$ of the whole pool of models can be compared to those of the selected pool.

* Kratky plot

As the strongest arguments for the flexibility of the N-terminus (which is a strong focus of this manuscript eg. see title) are the missing density in the x-tal structure and more importantly the scattering behavior in the Kratky plot (Figure 5a) I would expand a little bit more on this. Eg. the dimensionless nature of the plot could be highlighted in the text (multiplication of the q-vector (x -axis) by the particle's R_g and $I(q)$ by $(q \cdot R_g)^2$ instead of q^2). To emphasize the expected behavior of globular structures the location of a peak at $\sqrt{3}$ with a magnitude of $3 \cdot e^{-1}$ (1.104) can be indicated in graph (this holds for particles obeying Guinier's approximation) (citation of Durand et al. Volume 169, Issue 1, January 2010, Pages 45-53 (and practical approach see Kikhney & Svergun. 2015 /FEBS Letters 589 (2015) 2570–2577)

 with this the novice SAXS reader can understand why the slight shift in the peak is an important observation

* SAXS data

If SAXS data are important for strong points made in the manuscript (as here: flexibility of N-termini) it is good to present the data according to Trehwella et al 2017 (ActaCryst.(2017). D73, 710–728). I.e. a table summarizing the SAXS data and submission of the data to SASBDB.org should be considered.

Reviewer #3 (Remarks to the Author):

The manuscript entitled "A dual role in regulation and toxicity for the intrinsically disordered N-terminus of the toxin GraT" by Dr. Talavera and colleagues presents new data on the structure and function of the atypical GraTA toxin-antitoxin system (TA). The authors have determined crystal structures of the GraA antitoxin dimer, GraTA complex and GraA in complex with the operator DNA fragment. The authors find that, contrary to the previously characterized TA, GraA (the antitoxin) is fully folded and does not contain an intrinsically unfolded domain, while the N-terminus of GraT (the toxin) is intrinsically disordered and could not be traced. Further, they show that this disordered N-terminus is required for the toxicity and mRNA cleaving activity of GraT. As well, the disordered segment of GraT prevents binding of the GraA-GraT complex to DNA and is involved in the derepression of the graTA promoter. These data are certainly novel, intriguing and important to those studying TA. The resolved structures explain mechanistically some very unusual traits of the GraTA module, which behaves opposite to the "normal" TA in many sense. This manuscript presents technically sound and indisputable data that do not fit the established concepts about TA. Instead of unstable antitoxin and stable toxin, the authors describe a stable antitoxin and a toxin that contains a disordered part. The function of this unusual TA within the larger framework of bacterial life remains to be described.

The structure of the GraA-operator complex is intriguing as well. It has been previously determined that GraA binds specifically to the operator sequence that overlaps the transcriptional start site of the graTA promoter (REF 19, Tamman et al. 2014). The 35-nt DNA sequence that was protected from DNase I was a bit longer than the longest (33 bp) DNA fragment that was used to study the DNA-protein complex formation in this study. The authors demonstrate that two GraA dimers bind to the opposite faces of the double helix at the operator and interact the sugar-phosphate backbone. Based on the crystal structure, GraA does not make base-specific contacts and it remains unclear how this protein can recognize a unique operator and regulate transcription of a single operon. ITC measurements (in Table 1, Fig S3) showed, however, that some mutations in the central palindromic sequence of the operator fragment weaken GraA binding 100 to 10 000 times, while EMSA and SEC showed that the operator binding of two GraA dimers is highly cooperative. The DNA-bound GraA dimers do not interact each other and do not distort DNA. The authors suggest that the cooperativity is achieved by an accumulation of small DNA deformations and/or changes in DNA dynamics.

This MS certainly provides a lot of new information to the TA field.

It remains a question how much the results of the GraA DNA-binding experiments were affected by the length of the DNA fragment(s) used. Based on the resolved structure, the 33-bp fragment, which was used in most experiments, can accommodate exactly two GraA dimers on opposite faces of the helix.

Minor points:

As the pages and lines lack numbers, I start from the beginning.

Introduction.

"...rejuvenation of the toxin target (REF 8, 14)" remains unclear to a reader without an earlier knowledge. I suggest "dissociation of the complexes of a toxin (CcdB) and its target (gyrase)"
Results.

Subtitle: "Two GraA dimers are bound simultaneous on opposite sides" should be " ... are bound simultaneously on ..."

"This "specifically binding" HTH motif binds with..." is not correct because there are no base-specific contacts. Change to "This DNA-interacting HTH motif ..."

"... interact with two distal non-overlapping half-sites ..." In this sentence, "non-overlapping" is

confusing and should be deleted.

"we designed two different constructs of the GraT2A2 complex" change to "... two different constructs for expression of the GraT2A2 complex"

"In order to confirm that this is also true in *P. putida*"; mRNAse activity likely does not depend on an organism where it is tested. I recommend to change: "In order to confirm this result in *P. putida*"

"For the $\Delta 22\text{graT}$ -encoding strains, growth mirrors *P. putida* carrying the wild-type *graTA* operon, independent of the presence of the *graA* gene". A confusing sentence. Change to "The $\Delta 22\text{graT}$ -encoding strains grow just like the wild-type strain"

Paragraph: GraT disordered region is involved in derepression of the *graTA* promoter in vivo. Plasmid names in this paragraph make reading difficult. These are available in Materials and Methods and can be deleted from here.

Discussion

"intrinsic disorder is notoriously present in antitoxins" – why is intrinsic disorder notorious?

Materials and Methods

"Temperature tolerance tests" is a misleading subtitle as it recalls testing survival at extreme temperatures. I suggest "Temperature-sensitive growth assay"

Table 1; for clarity, the "full palindrome" and "half palindrome" sequences should be underlined or in bold. In the wt sequence, the contact sites of GraA residues Arg46 and Asn42 with the DNA backbone should be indicated

Fig 6; instead of "empty pBBR" use "empty vector"

Fig 7 legend; instead of "GraT N-terminal flexibility..." use "The flexible N-terminus of GraT..."

Fig 8 legend; I suggest "The flexible N-terminus of GraT is required for derepression of the *graTA* promoter"

Fig 8A is not necessary; it contains trivial information and is difficult to follow because of the use of plasmid ID-s and tiny font. Explanation of such a simple experiment does not require a scheme.

Fig 8B and C – change pBBRlactac to "empty vector" for clarity. What do the error bars show in Fig 8B? Fig 8C contains one empty lane on the left (mw marker?) without any title.

Fig S3 legend - the reference that the variants are from Table 1 is missing. Which sequences are boxed?

The table of bacterial strains, plasmids and oligonucleotides in the supplement requires a different title. Probably Table S2?

Reviewer #1 (Remarks to the Author):

The manuscript submitted by Talavera et al. reports a structural and functional studies on the GraA GraT antitoxin toxin system. The authors describe their experimental evidence for GraA binding to the promoter DNA from which they evolve a model for cooperative binding of two GraA-dimers on opposing sides of the palindromic sequence within the GraA promoter region. Interestingly, no direct specific readout of the DNA sequence is established as not a single contact between residues of GraA and any base moiety is observed in the Gra2/DNA crystal structure. Thus, the authors speculate about a kind of allosteric signaling mediated by the DNA resulting in the observed cooperative binding. Furthermore, the authors convincingly show that a single HTH-motif of each GraA dimer binds to the central palindromic sequence and confers specificity. In contrast, the second HTH-motifs seems to contribute only to affinity but not to specificity for the DNA. From their structural and functional studies, the authors infer that the N-terminal 21 residues of the GraT toxin impair with promoter binding. Furthermore, the authors show that removal of this region results in a non-functional toxin.

The manuscript is well written in most parts and comprehensible. However, as discussed in detail below, there are remaining unclear but important issues.

Comment 1: Title and abstract: As outlined below in detail in comment 5, it remains to be shown whether the N-terminus of GraT is indeed intrinsically disordered or whether this region is just unstructured in the GraAT complex. Furthermore, most parts of the manuscript describe the structural and functional studies on GraA promoter binding. Thus, the title might be misleading and should be revised

Authors: To address this problem, it was necessary to produce GraT in absence of GraA, which was problematic and is the main reason why our revision took such a long time. Initially we tried to isolate GraT from the GraTA complex via unfolding/refolding protocols, but the resulting protein N-terminally His-tagged GraT could not be eluted from the Ni-sepharose matrix. We then tried a number of different constructs with the histidine tag placed at different positions of GraT or GraA and with or without GraA, each of which gave either problems with expression, refolding (where required) or with purification. We finally succeeded using a construct where The C-terminal His92 is replaced by a TEV cleavage site followed by a histidine tag (ENLYFQGSAGHHHHH). The His-tag could be successfully cleaved off and the corresponding SAXS data are included as panels C and D in Figure 5. The normalized Kratky plot here is indicative of the presence of a significant amount of intrinsic disorder, and this is further confirmed by comparing the theoretical SAXS curves of fully folded and disorder-modelled GraT with the experimental data. Thus we conclude that also in its isolated state, GraT contains a significant amount of intrinsic disorder, justifying our original statements.

Comment 2: Nature of the DNA helix in the crystal contacts. It remains unclear from the manuscript, whether the 33 bp of dsDNA forms an undisturbed entity with the crystal packing or something like a continuous helix as observed in several other protein-DNA co-crystal structures is formed. Can the authors exclude crystal-packing artifacts to cause the lack of structural features (be it bending / straightening or different rigidity) that would allow to explain sequence specificity?

Authors: Indeed the crystal packing is achieved by the formation of continuous DNA helices in the directions of the crystallographic a and b axes (Response_Figure_1_A,B). Nevertheless the SAXS data of the GraA-DNA complex show that the conformation of the complex in the crystal nicely fits the structure of the complex in solution (Response_Figure_1_C,D and Figure_S2 A,B). This is now included in the text and reads like: "The fitting of the theoretical scattering solution of atomic the coordinates of GraA-DNA complex into the experimental scattering curve ($\chi^2 = 4.6$) (Figure. S2 A,B) corroborates that this is also the conformation of the complex in solution.

Response Figure 1. Crystal packing of the GraA-DNA complex. A. Crystal lattice formed by the GraA2-DNA complex. The c axis of the unit cell is oriented perpendicular to the figure and to the a and b axes (in green) are at the same

plane of the figure. The crystal contacts along the a and b axes create a “continuous” DNA helix. **B.** Simplified view of panel A where the GraA2 was removed for a better visualization of the “continuous” DNA helix. **C.** Fitting of the theoretical (red line) and experimental (open circles) saxs curves of the GraA-DNA complex. The theoretical scattering curve was calculated using the crystal structure of the GraA-DNA complex, showing that it is in agreement with the structure of the complex in solution. **D.** Ab-initio envelope of the complex calculated from the experimental SAXS data is shown as a gray meshed surface.

Comment 3: Table 1 and Figure S3. Since the numbers are quite high, it is likely that those are presumably not Kd but KA values. I guess the latter is correct as the authors stated in the Materials and Methods section that KA values were calculated. For all ITC experiments the quality of the fit cannot be judged, and no errors are given. Additionally, has the heat of dilution been considered in ITC experiments. This is important to some extent, as the fitted entropic and enthalpic contributions are given as well.

Authors: The errors are now included in the table. We removed the ΔS column because the error estimation are too high. This is not unusual because the error estimates for ΔS correspond to the sum of the errors from the two fitted parameters ΔH and ΔG (K_a). This issue becomes more evident when the binding gets weaker and the estimations of the fitted parameters ΔH and K_a become more inaccurate. And yes, for each titration the heat of dilution was subtracted. This is now added to the Material and methods section: "*Raw data were integrated and corrected for the buffer dilution heat effects*".

Comment 4: Architecture of the GraAT protein complex. It would be interesting if the authors could give some information how GraA inhibits GraT activity.

Authors: We added: "*Based on extrapolation from the structure of HigB in complex with the ribosome²⁹ (PDB entry 4W4G), the active site of GraT, formed by the cleft around His92, is not occluded by GraA. Instead, GraA inactivates GraT by inhibiting its binding to the ribosome due to steric hindrance via clashes between GraA and the ribosomal RNA as well as ribosomal protein S13 (Fig, 5 E).*"

Comment 5: The authors unambiguously show that the N-terminus of GraT is disordered in the GraAT complex. However, it remains to be shown whether this part of the toxin is ordered in GraT in the absence of GraA. Most importantly, it remains unclear, whether truncation of the latter in GraT makes the protein in-stable. When judging the protein levels shown in Figure 8 panel C, much less protein is detected in *E. coli* when compared to the full-length protein (excluding technical issues such as for instance different affinity of the polyclonal GraT-antibody for the two variants or that antibody binding regions have been removed). Is this variant less stable or less expressed in this experiment? Moreover, the authors speculate that the N-terminal region of

GraT is too distal to the catalytic site to have any direct impact on activity. It would be interesting to know, what the distance between the active site and the N-terminus is. In fact, the extended model for the N-terminus shown in Figure 5 panel B would suggest that this part could wrap around the entire surface of the molecule and reach any point at the molecular surface of GraT. Finally, could thermostability relate to the observed cold-sensitive phenotype in *P. putida* and has the truncation any effect on that?

Authors: As already mentioned in our response to comment 1, we finally succeeded in preparing a tagless GraT in absence of GraA that is sufficiently soluble and can be produced in sufficient amounts to perform SAXS experiments. These experiments on this isolated GraT show the presence of intrinsic disorder in GraT in absence of GraA and thus answer what we agree was the most crucial shortcoming in our original manuscript.

Whether deletion of the N-terminus of GraT changes its *in vivo* lifetime is indeed another major issue. To answer this question, we made use of the GraT-1C mutant. The latter is a nontoxic variant of GraT which lacks the C-terminal histidine residue. This way we can compare the two proteins ($\Delta 22$ GraT and GraT-1C) in absence of GraA without having to deal with effects of toxicity of the wild-type protein. Both proteins were expressed in *E. coli* at 30°C and after inhibition of translation by chloramphenicol, the samples were taken and analyzed on SDS PAGE and by Western blot. Our results show that both proteins are highly stable for at least 4 hours. This indicates that the N-terminal deletion does not affect the lifetime of GraT in a biologically relevant way the protein.

It may be that the presence of the N-terminus affects the thermodynamic stability of the protein, which could be related to the cold-sensitive phenotype. Unfortunately, at the concentrations where we might be able to measure this *in vitro*, the $\Delta 22$ GraT variant behaves as a tetramer, and therefore temperature-induced unfolding of GraT and $\Delta 22$ GraT by for example followed by CD spectroscopy cannot be directly compared. However, we can compare the GraTA and $\Delta 22$ GraTA complexes which behave identical (Response Figure 2). Within the physiologically relevant temperature range of 5 - 40°C, both complexes remain well-folded, indicating again that the N-terminal segment of GraT does not affect conformation or overall thermodynamic stability in a biologically relevant way.

One should also remember that GraT toxicity is enhanced by the major chaperone DnaK (Ainelo *et al.*, 2016), which on the one hand indicates that folding is an issue but also again complicates interpretation of *in vitro* experiments on thermodynamic stability (see also our response to Comment 6).

Response Figure 2: Secondary structure content of GraTA and $\Delta 22$ GraTA: To determine the temperature dependency of the secondary structure content of GraTA and $\Delta 22$ GraTA we measured the circular dichroism spectra of GraTA and $\Delta 22$ GraTA GraT from 5 to 40 0 °C.

Residue 23 (just after the unseen N-terminal 22 amino acids IDP region) is located 23 Å away from His92. Thus in theory this N-terminus would be able to reach the active site, but we think it is unlikely to contribute directly to catalysis. In the discussion it now reads: *“The N-terminal 22 amino acids segment of GraT toxin not only affects operator binding, but also the activity of the protein. The latter is unexpected given that the predicted active site near His92 is very similar to the active site of HigB and the N-terminus does not likely contribute catalytically important residues. While the 22 N-terminal residues are flexible and therefore might be considered to move towards the active site, this would likely prevent productive binding to the ribosome.*

*Therefore it is more likely that upon ribosome binding this region would fold in such a way that it facilitates the interaction between the ribosome and GraT without directly being involved in catalysis. In line with that, the crystal structure of *P. vulgaris* HigB bound to the ribosomal 30S subunit demonstrated that the first N-terminal α -helix of HigB, the very same region that is unstructured in GraT, is important for HigB recognition of the 16S rRNA in the ribosomal A site²⁹. Given that several substitutions in HigB $\alpha 1$ decrease HigB toxicity by disturbing its ribosome binding²⁹, it is reasonable to assume that the N-terminus of GraT also plays a role in ribosome binding.”*

The issue of different levels of expression between GraT and $\Delta 22$ GraT is discussed in our answer to comment 7.

Comment 6: Along the same line as outlined in comment 5: Given the homology with for instance HigB, it is convincing that the loss of ribosome binding could be the reason for the lack of activity when N-terminal truncated variant of GraT were used in lpp mRNA degradation experiments. Yet, can the authors provide any additional evidence for their hypothesis?

Authors: Given that GraT is a ribosome-dependent mRNAse (Figure 6), it definitely should interact with the ribosome. Unfortunately, despite several efforts we could not detect GraT on ribosomes purified from an antitoxin-deficient *Pseudomonas* strain. This may indicate that GraT binding to ribosome is either transient or so weak that is lost during the ribosome purification process. Contrary to that, HigB association with ribosomes was reported to be easily detectable (Hurley & Woychik, 2009). Intriguingly, previous research revealed that the DnaK chaperone binds GraT and is somehow involved in the GraT-caused growth inhibition and ribosome biogenesis defect (Ainelo *et al.*, 2016). Our unpublished data suggest that DnaK facilitates GraT toxicity. How DnaK does this is currently not known, but it can be conceived that our lack of detection of ribosome-GraT complexes *in vitro* may have to do with this factor.

Comment 7: De-repression of the operon by GraT. Apparently, the protein levels of full-length and truncated GraT are very different in E. coli (excluding issues with the antibody used for the WB shown in Figure 8 panel C). Thus, the observed transcriptional repression of the operon upon expression of the N-terminal truncated GraT in the beta galactosidase assay could be mainly caused by GraA alone which is in huge excess over the truncated GraAT complex. Can the author exclude this?

Authors: The stability assay of GraT derivatives also allowed us to evaluate the affinity of the anti-GraT antibodies to $\Delta 22$ GraT and GraT-1C. Both GraT variants were similarly detected by anti-GraT antibodies. Thus, the similar stability and the similar detection of GraT-1C (expectedly also full-length GraT) and $\Delta 22$ GraT indeed suggests, as the reviewer indicated, that the $\Delta 22$ GraT is relatively less expressed than GraT. However, it does not necessarily mean that in case of $\Delta 22$ GraTA complex, the level of $\Delta 22$ GraT is lower than that of GraA. Comparison of two western blots assayed with two different antibodies does not allow drawing conclusions about stoichiometry of GraA and GraT. Importantly, to increase the GraT expression with respect to GraA we replaced the native SD of the *graT* in front of the *graTA* operon with more efficient one (this is now mentioned in the last paragraph of Results, page 10). In case of *tac-graTA* cassette, this results in higher expression of GraT than of GraA (*graA* is preceded by native SD), as indicated by 1) full derepression of *graTA* promoter by GraTA complex (Figure 8), 2) GraT mRNAse activity when GraTA complex was expressed at 20°C (Figure 6; note, that the very same plasmid with *tac-graTA* cassette was used in both mRNAse and β -gal assays). Particularly the latter result shows that GraT is in excess; otherwise it would not be toxic. In case of *tac- $\Delta 22$ graTA* cassette we, unfortunately, do not have good measure of $\Delta 22$ GraT and GraA stoichiometry. Yes, western blot indicates that, compared to GraTA complex, the stoichiometry of two proteins is different and more declined in favour of GraA, but we cannot say that the GraA “is in huge excess over the truncated GraAT complex”. Given, that in case of *tac-graTA* cassette GraT is in excess (GraT>GraA), this decline in case of *tac- $\Delta 22$ graTA* cassette could mean either the stoichiometry of $\Delta 22$ GraT>GraA or $\Delta 22$ GraT=GraA or $\Delta 22$ GraT<GraA.

Taken together, as we cannot measure the GraT/GraA ratio in in vivo β -gal experiment (expression of GraTA proteins in β -gal assay is too low to see them in cell lysates when just analysed in SDS PAAG), we cannot rebut the referees' concern that "the observed transcriptional repression of the operon upon expression of the N-terminal truncated GraT in the beta galactosidase assay could be mainly caused by GraA alone" (comment 7). However, even if this is so then our main point, the de-repression of the promoter by the wild-type GraTA complex, is clearly supported by in vivo β -gal data. To emphasize this, we essentially rewrite the in vivo promoter analysis paragraph and changed also its title (pages 10-11).

Comment 8: Discussion: ...This region of GraT remains disordered when bound to the antitoxin... As the authors do not have any experimental evidence for that, it is advisable toning this statement down.

Authors: We have now also been able to provide data that also in its isolated state GraT contains an intrinsically disordered segment. We believe that our SXAS data on GraT and GraTA as well as the crystal structure of the GraTA complex provide sufficient evidence for the presence of intrinsic disorder. We therefore now state that "*The 22 N-terminal residues of GraT are disordered in both the isolated state as well as in the GraTA complex and prevent the TA complex from interacting with the operator*".

Comment 9: Similarly, ...the 22-amino acid intrinsically disordered sequence at the N-terminus seems to be solely responsible for the inability of the GraT-GraA complex to bind its target DNA sequence... and the entire residual paragraph do require some reconsideration unless further evidence supporting this hypothesis is provided.

Authors: We agree to tone down this sentence: "... the 22-amino acid intrinsically disordered sequence at the N-terminus seems to be a major contributor to the inability of the GraT-GraA complex to bind its target DNA sequence". We nevertheless stick to the remainder of the paragraph, which we do not think is overstated. ITC analyses and EMSA data unambiguously show that the N-terminal disordered region of toxin GraT inhibits the binding of GraTA complex with DNA. When this region is deleted, the GraTA complex binds DNA with high affinity (Figure 7), indicating that particularly the N-terminus of GraT is implicated in "non-binding" phenotype of GraTA. To remind that in the discussion, we added a reference to Figure 7 (page 12). In vivo data is in full accordance with the ITC and EMSA results as full-length GraTA complex cannot inhibit (and thus bind to) the *graTA* promoter (Figure 8). In the case of truncated GraTA complex the *graTA* promoter is repressed like in case of GraA. The reviewer is right that the latter result (the truncated GraTA phenotype=GraA phenotype) raises the question whether the promoter repression is caused by GraA only or by truncated GraTA complex or by both (ITC measurements showed that the truncated GraTA binds DNA even better than GraA). However, this possibility does not, in any way, discredit the conclusion drawn from *in vitro* ITC and

EMSA data, that the N-terminus of GraT prevents the binding of truncated GraT complex.

Comment 10: ...The DNA distortions that affect the binding koff are dampened ... should rather read like ... The DNA distortions that affect the dissociation kinetics are dampened ... or something like that.

Authors: We agree and changed this to "*The DNA distortions that affect the dissociation kinetics are dampened*"

Comment 11: The entire paragraph starting with ... Intrinsic disorder in the GraT toxin not only affect operator binding ... might require some reconsideration based on the lack of information about the structure GraT alone or when bound to the ribosome.

Authors: This paragraph now reads: "*The N-terminal 22 amino acids segment of GraT toxin not only affects operator binding, but also the activity of the protein. The latter is unexpected given that the predicted active site near His92 is very similar to the active site of HigB and the N-terminus does not likely contribute catalytically important residues. While the 22 N-terminal residues are flexible and therefore might be considered to move towards the active site, this would likely prevent productive binding to the ribosome.*

Therefore it is more likely that upon ribosome binding this region would fold in such a way that it facilitates the interaction between the ribosome and GraT without directly being involved in catalysis. In line with that, the crystal structure of P. vulgaris HigB bound to the ribosomal 30S subunit demonstrated that the first N-terminal α -helix of HigB, the very same region that is unstructured in GraT, is important for HigB recognition of the 16S rRNA in the ribosomal A site²⁹. Given that several substitutions in HigB α 1 decrease HigB toxicity by disturbing its ribosome binding²⁹, it is reasonable to assume that the N-terminus of GraT also plays a role in ribosome binding."

Comment 12: It was somewhat unclear from the main text that translation of GraT is initiated from an individual SD sequence. It might be worthwhile to briefly mention that fact somewhere in the manuscript.

Authors: To achieve the higher expression of GraT over the GraA, the SD sequence in front of the *graT* gene (in both *tac-graTA* and *tac- Δ 22*graTA* cassettes) was replaced with an effective SD from the pET11c plasmid. This is now specified also in the main text (page 10): "*For enabling expression of GraT at a higher level than GraA, the native SD sequence in front of *graT* was replaced with strong SD from the pET11c expression plasmid.*"*

Comment 13: Figure 1 panel B: What were the limits for the electrostatic surface potential?

Authors: 10 kT blue, 0 white, -10kT red. Now it is inserted in the figure legend.

Comment 14: Figure 3: What were the experimental conditions for this experiment? The authors might want to mention briefly which column they have used for size exclusion chromatography.

Authors: We used a 70 kDa cut-off BioRad SEC70 column. It is now inserted in the figure legend.

Comment 15: Figure 7 panel A and B: Are the given constants association or dissociation constants?

Authors: This figure is now corrected and shows the association constant K_A .

Comment 16: Table S1: The formatting of the space group symbols is not correct and ambiguous (P32 1 1 or P3 2 1. Furthermore, the PDB codes given in table do not match with the file names of the deposited validation reports.

Authors: We corrected the space group symbols to: $P2_1$, $P3_2$ and $P4_1$. The PDB entry code is now fixed as well (6F1X instead of 6FIX in table S1).

Comment 17: As the criteria for chosen resolution cutoff of the low-resolution structure of the GraA2/DNA complex is very different when compared to the two other structures, it would be worth to show a representative electron density map demonstrating the quality of the map.

Authors: This figure is now included in the manuscript as supplementary Figure S2 A,B.

Reviewer #2 (Remarks to the Author)

A dual role in regulation and toxicity for the intrinsically disordered N-terminus of the toxin GraT

Talavera et al. have thoroughly characterized a bacterial toxin-antitoxin system with a wide range of biochemical and biophysical methods. The manuscript is clearly written and the results as well as the conclusions are well presented. The findings are interesting also for a wider audience

Some comments regarding the structural properties of the GraT N-terminus:

*** Length of 22 residues for the intrinsically disordered N-terminus;**

- Are the other intrinsically disordered regions (observed for other anti-toxins) of similar length?

Authors: In most cases, the disordered region in the antitoxin corresponds to about 50% of the sequence. E.g. for F-plasmid CcdA this is 36 aa out of 72; for *E. coli* MazE its is 38 out of 82 aa. For Phd it is also 23 out of 73 aa, thus the same length as for GraT (but larger in %). Thus the IDP stretch of GraT is not unusual in terms of length, but is relatively short compared to the full length of the GraT protein (22 aa out of 92 - 24%).

*** SAXS models as shown in Figure 5B:**

- in the orientation of the upper structure it looks as if the extensions of the blue models are at the C-terminus (extension of the helix) as opposed to the N-terminus (compare to the orientation in Figure 4)

Authors: This was indeed a mistake in figure 4. Now it is corrected.

- selected ensemble: the ensemble shown here overlap pretty much. Thus, the flexibility seems rather constrained. Does the program MultiFOXS address this? Eg, the distribution of Dmax/Rg/Volume of the whole pool of models can be compared to those of the selected pool.

Authors: As is now described in the text, MultiFoxs was used to generate a large ensemble of 10.000 conformations, which should be enough to cover the required conformational space to select from.

Thus we now write: "*At first glance the dimensionless Kratky plot of GraT2A2 shows that is globular (Fig. 5A). Nevertheless as the predicted disorder part is only 11 % of the full complex and might not be captured by the SAXS experiment, we wanted to know which possible conformation of GraT2A2 best fits the experimental scattering data: a model with a disordered N-terminus or a model with the N-terminus folded as in P. vulgaris HigB. For the former option we used the online program MultiFoxs33. By defining the first 22 amino acids of GraT as disordered (flexible), this software generated 10 000 conformations from where it selected a three-states model as the best fit to the experimental scattering curve with a $\chi^2 = 1.4$ (Fig. 5B (blue curve)). This three-states model shows an Rg distribution with one major peak from 21.5 to*

23.5 Å with weight of 0.86 (Fig. S6). For the latter option we modeled the 22 N-terminal amino acids in the conformation displayed by the corresponding segment of *P. vulgaris* HigB and kept the remainder of GraT as in the GraT2A2 crystal structure, and again compared this model to the experimental scattering curve (Figure 5 (red curve)). In this case the fitting of the theoretical to the experimental scattering curve was worse with a $\chi^2 = 10.5$. All this together, suggests that, despite the shape of the dimensionless Kratky plot, the N-terminal segment of GraT might indeed be disordered but is not picked in this plot because this region only represents the 10% of the protein."

* Kratky plot

As the strongest arguments for the flexibility of the N-terminus (which is a strong focus of this manuscript eg. see title) are the missing density in the x-tal structure and more importantly the scattering behavior in the Kratky plot (Figure 5a) I would expand a little bit more on this. Eg. the dimensionless nature of the plot could be highlighted in the text (multiplication of the q-vector (x-axis) by the particle's R_g and $I(q)$ by $(qR_g)^2$ instead of q^2). To emphasize the expected behavior of globular structures the location of a peak at $\sqrt{3}$ with a magnitude of $3 \cdot e^{-1}$ (1.104) can be indicated in graph (this holds for particles obeying Guinier's approximation) ³¹ citation of Durand et al. Volume 169, Issue 1, January 2010, Pages 45-53 (and practical approach see Kikhney & Svergun. 2015 /FEBS Letters 589 (2015) 2570–2577)

 with this the novice SAXS reader can understand why the slight shift in the peak is an important observation

A detailed description of the dimensionless Kratky plot is now included in the results section: "*The use of the dimensionless Kratky plot $((qR_g)^2 I(q)/I(0)$ vs qR_g) is a relatively easy way to show that a protein is completely folded, partially folded or completely unstructured³¹. If a protein is globular it follows Guinier's law $I(q)/I(0) = \exp(-(qR_g)^2/3)$. The corresponding dimensionless Kratky plot is a function $f(x) = x^2 \exp(-x^2/3)$, with $x = qR_g > 0$ with maximum of 1.104 at $qR_g = \sqrt{3}$. On the other hand an ideally disordered protein follows Debye's law $I(q)/I(0) = 2(x^2 - 1 - \exp(-x^2))/x^4$, with $x = qR_g > 0$. In this case the Kratky plot is described by the function $f(x) = 2(x^2 - 1 - \exp(-x^2))/x^2$ which increases monotonically with an asymptote at $f(x) = 2$. Experimentally, globular proteins show a very similar normalized Kratky plot with a maximum at $(\sqrt{3}; 1.1)$, while partially unstructured proteins show a maximum shifted to higher values in both axes³².*"

Furthermore, we modified figure 5. Now it includes the dimensionless Kratky plots (panels A and C) and the fitting of the theoretical scattering curves, considering GraT ordered or disordered, to the experimental scattering obtained for the GaTA complex (panel B) or isolated GraT (panel D), respectively. We also cite the two suggested references (here refs 31 and 32).

Most importantly, this comment from the reviewer made us re-assess our SAXS data, and we concluded that the quality might not be good enough to warrant the

conclusions we drew. Thus we spend time and effort to re-measured the data at higher concentrations. These new data are now presented according to the guidelines described in Trehwella et al. in Supplementary Table S2.

The Dimensionless Kratky plot does not show unambiguously the presence of large amounts of disorder. Although we believe that in the GraTA complex the N-terminus of GraT remains disordered (as witnessed in the crystal structure), the amount of disorder (22 aa out of 92 + 99 = only 11%) is arguably too small to be observed in this way. Nevertheless, comparing the theoretical scattering curves of N-terminally disordered GraTA, fully folded GraTA (as in HigBA) and N-terminally truncated GraTA with the experimental data gives a strong indication that the disordered conformation is still present in the complex (Figure 5B).

On top of that we added the new SAXS data for GraT in its isolated state, which unambiguously shows the presence of intrinsic disorder via both the dimensionless Kratky plot and model-based prediction of the scattering curves. These data are now also presented in a tabulated form as requested. This is added as:

"To corroborate the previous hypothesis we expressed and purified an inactive form of the toxin. This form consists of the substitution of the C-terminal H92 by a TEV cleavage site followed by a histidine tag (ENLYFQGSAGHHHHH). After proteolysis with TEV this new version of GraT is only five amino acids longer than the wild type (< 6% difference). In this case, the dimensionless Kratky plot (Fig. 5C) shows a clear proof of disorder with a shift of the maximum towards higher values (2.0; 1.22) with respect to the maximum for globular proteins with coordinates (1.73; 1.1). Furthermore, the theoretical scattering curve of the disordered ensemble fits much better the experimental scattering curve than the HigB-like conformation (Fig. 5D), with $\chi^2 = 3.1$ and 8.9, respectively. In this case the ensemble consists of a four state model with a major peak in the R_g distribution from 13 to 15 Å with a weight of 0.8 and three other small peaks at larger R_g 's with weights smaller than 0.1 (Fig. S3 B)."

*** SAXS data**

If SAXS data are important for strong points made in the manuscript (as here: flexibility of N-termini) it is good to present the data according to Trehwella et al 2017 (ActaCryst.(2017). D73, 710–728). I.e. a table summarizing the SAXS data and submission of the data to SASBDB.org should be considered.

As already mentioned, all SAXS data are now presented according to the guidelines described in Trehwella et al. in Supplementary Table S2. As requested, the SAXS data have now been submitted at SASBDB.org with accession numbers: SASDE68 (GraA-Operator complex), SASDE58 (GraTA complex), SASDE48 (GraT).

Reviewer #3 (Remarks to the Author):

The manuscript entitled "A dual role in regulation and toxicity for the intrinsically disordered N-terminus of the toxin GraT" by Dr. Talavera and colleagues presents new data on the structure and function of the atypical GraTA toxin-antitoxin system (TA). The authors have determined crystal structures of the GraA antitoxin dimer, GraTA complex and GraA in complex with the operator DNA fragment. The authors find that, contrary to the previously characterized TA, GraA (the antitoxin) is fully folded and does not contain an intrinsically unfolded domain, while the N-terminus of GraT (the toxin) is intrinsically disordered and could not be traced. Further, they show that this disordered N-terminus is required for the toxicity and mRNA cleaving activity of GraT. As well, the disordered segment of GraT prevents binding of the GraA-GraT complex to DNA and is involved in the derepression of the graTA promoter. These data are certainly novel, intriguing and important to those studying TA.

The resolved structures explain mechanistically some very unusual traits of the GraTA module, which behaves opposite to the "normal" TA in many sense. This manuscript presents technically sound and indisputable data that do not fit the established concepts about TA. Instead of unstable antitoxin and stable toxin, the authors describe a stable antitoxin and a toxin that contains a disordered part. The function of this unusual TA within the larger framework of bacterial life remains to be described.

The structure of the GraA-operator complex is intriguing as well. It has been previously determined that GraA binds specifically to the operator sequence that overlaps the transcriptional start site of the graTA promoter (REF 19, Tamman et al. 2014). The 35-nt DNA sequence that was protected from DNase I was a bit longer than the longest (33 bp) DNA fragment that was used to study the DNA-protein complex formation in this study. The authors demonstrate that two GraA dimers bind to the opposite faces of the double helix at the operator and interact the sugar-phosphate backbone. Based on the crystal structure, GraA does not make base-specific contacts and it remains unclear how this protein can recognize a unique operator and regulate transcription of a single operon. ITC measurements (in Table 1, Fig S3) showed, however, that some mutations in the central palindromic sequence of the operator fragment weaken GraA binding 100 to 10 000 times, while EMSA and SEC showed that the

operator binding of two GraA dimers is highly cooperative. The DNA-bound GraA dimers do not interact each other and do not distort DNA. The authors suggest that the cooperativity is achieved by an accumulation of small DNA deformations and/or changes in DNA dynamics.

This MS certainly provides a lot of new information to the TA field.

We are happy with the positive reception of our manuscript by this referee.

It remains a question how much the results of the GraA DNA-binding experiments were affected by the length of the DNA fragment(s) used. Based on the resolved structure, the 33-bp fragment, which was used in most experiments, can accommodate exactly two GraA dimers on opposite faces of the helix.

The 33 bp fragment was identified in an earlier study (Tamman et al., 2014 J. Bacteriol. 196, 157-169) and according to our crystal structure does extend a few bases beyond the exact binding sites of GraA. Shorter fragments do not bind, but indeed we did not test longer fragments. It is unlikely though that further extending the DNA fragment on either site will significantly affect the results. Affinities may become slightly higher due to better stabilization of the DNA duplex (although the 33 bp duplex is not expected to dissociate at room temperature). It is difficult to conceive that a longer piece of DNA would qualitatively change the effect of the N-terminus of GraT given that the possibilities for steric hindrance can only increase.

Minor points: As the pages and lines lack numbers, I start from the beginning. Introduction. "...rejuvenation of the toxin target (REF 8, 14)" remains unclear to a reader without an earlier knowledge. I suggest "dissociation of the complexes of a toxin (CcdB) and its target (gyrase)"

Authors: The complete sentence now reads: "*Most type II antitoxins contain an intrinsically disordered region that is required not only for neutralizing the toxin and forming the repressor complex^{8,22} but also for its rapid degradation²³ and for the dissociation of the toxin from its target (e.g. CcdB and Gyrase)^{8,14}*"

Results. Subtitle: "Two GraA dimers are bound simultaneous on opposite sides" should be "... are bound simultaneously on ..."

Authors: This is corrected

"This "specifically binding" HTH motif binds with..." is not correct because there are no base-specific contacts. Change to "This DNA-interacting HTH motif ..."

Authors: This is now rephrased as: "*This "central binding" HTH motif*"

"... interact with two distal non-overlapping half-sites ..." In this sentence, "non-overlapping" is confusing and should be deleted.

Authors: we accepted this suggestion

"we designed two different constructs of the GraT₂A₂ complex" change to "... two different constructs for expression of the GraT₂A₂ complex"

Authors: This is changed to: "*we designed two different constructs for the expression of the GraT₂A₂ complex*"

“In order to confirm that this is also true in *P. putida*”; mRNAse activity likely does not depend on an organism where it is tested. I recommend to change: “In order to confirm this result in *P. putida*”

Authors: This is changed as requested

“For the $\Delta 22$ graT-encoding strains, growth mirrors *P. putida* carrying the wild-type graTA operon, independent of the presence of the graA gene”. A confusing sentence. Change to “The $\Delta 22$ graT-encoding strains grow just like the wild-type strain”

Authors: We rephrased this as: “*The $\Delta 22$ graT-encoding strains grow just like the wild-type *P. putida* strain, independent of the presence of the graA gene*”.

Paragraph: GraT disordered region is involved in derepression of the graTA promoter in vivo. Plasmid names in this paragraph make reading difficult. These are available in Materials and Methods and can be deleted from here.

Authors: This paragraph has been completely re-written and now reads:

“Overexpression of the GraTA complex results in derepression of the graTA promoter in vivo

The role of GraT in derepression of the graTA promoter was further tested in vivo. The graT-lacZ transcriptional fusion was used as a reporter for graTA promoter activity in E. coli. The genes coding for GraTA, GraA or $\Delta 22$ GraTA were cloned under an IPTG-inducible tac promoter. For enabling expression of GraT at a higher level than GraA, the native SD sequence in front of graT was replaced with strong SD from the pET11c expression plasmid. Note that for expression of the GraTA complex, the same plasmid was used as in mRNAse assay. The plasmids were independently transformed into E. coli already containing the graT-lacZ reporter (Figure 8A). β -galactosidase measurements show that while GraA alone represses the graTA promoter activity, the expression of the GraTA complex leads to full derepression of the promoter (Figure 8B). Expression of the $\Delta 22$ GraTA with N-terminally truncated GraT again represses the promoter (Figure 8B) which is in good accordance with strong binding of $\Delta 22$ GraT₂A₂ to the operator (Figure 7). Expression of the antitoxin and toxin proteins was verified by Western blot (Figure 8C). These results confirm the in vitro data that the GraT N-terminal segment is hindering the binding of GraTA complex to the operator and that GraT acts as a derepressor of the graTA operon transcription.”

Discussion: “intrinsic disorder is notoriously present in antitoxins” – why is intrinsic disorder notorious?

Authors: we changed "notoriously" to "abundantly".

Material and Methods: “Temperature tolerance tests” is a misleading subtitle as it recalls testing survival at extreme temperatures. I suggest “Temperature-sensitive growth assay”

Authors: We accept the suggestion of the referee and now use “Temperature-sensitive growth assay”

Materials and Methods: Table 1; for clarity, the “full palindrome” and “half palindrome” sequences should be underlined or in bold. In the wt sequence, the contact sites of GraA residues Arg46 and Asn42 with the DNA backbone should be indicated

Authors: In Table 1 the position of the palindromic sequence is now underlined in each DNA species.

Fig 6; instead of “empty pBBR” use “empty vector”

We changed this to "empty vector (pBBRLactac)"

Fig 7 legend; instead of “GraT N-terminal flexibility...” use “The flexible N-terminus of GraT...”

This is changed as requested.

Fig 8 legend; I suggest “The flexible N-terminus of GraT is required for derepression of the graTA promoter”

This is changed as requested

Fig 8A is not necessary; it contains trivial information and is difficult to follow because of the use of plasmid ID-s and tiny font. Explanation of such a simple experiment does not require a scheme.

We removed the schematic as requested

Fig 8B and C – change pBBRLactac to “empty vector” for clarity. What do the error bars show in Fig 8B? Fig 8C contains one empty lane on the left (mw marker?) without any title.

Authors: We replaced "pBBRLactac" by "empty vector (pBBRLactac)". The error bars indicate 95% confidence intervals of at least three independent experiments. This is now specified in the figure legend (page 28).

Fig S3 legend - the reference that the variants are from Table 1 is missing. Which sequences are boxed

The legend of this figure, which now has been renamed Figure S4, now reads: ***“Figure S4. Operator binding by GraA₂ in vitro. ITC experiments of different variants of the graTA operator (as in Table 1) titrated into GraA₂ at 25°C. Wild-type sequences are given in capital letters, while mutated base positions are given by small letters. The box represents the position of the central palindrome. For each panel, the upper part shows the heat generated by the individual injections while the lower part provides the corresponding fitted titration isotherm at 25°C.”***

The table of bacterial strains, plasmids and oligonucleotides in the supplement requires a different title. Probably Table S2?

Authors: this is corrected. It is now Supplementary table S2

REVIEWERS' COMMENTS:

Reviewer #2 (Remarks to the Author):

The Authors have gone to quite some length to try to collect further evidence that the N-terminus of GraT is indeed flexible. I still do not quite understand why this point has to be emphasized so strongly. As also suggested by Referee 1, the focus (of eg. the title) could be more on the promotor binding instead. All the other data is nicely presented, the MS well written and a interesting topic. As stated below I would suggest down playing the "flexibility".

- The SAXS data have been made available upon request (also corresponding tables have been added to the MS and the data uploaded to the SASBDB database)

* it would be nice if the models could later be added to database as well.

* the section on Kratky plot (lines 230 -239) could be placed in Material and Methods

Some questions arise regarding the SAXS data:

* SEC-SAXS data collection has been performed. It would be good to state how $I(0)$ values were derived from here, ie. how was the concentration measured and taken into account? Or were other concentration independent MW estimates made.

*Note, R_g is derived from the Guinier Plot (which also delivers $I(0)$), please correct: " R_g is derived from $I(0)$ " in the material and method section

* As DNA and protein have different contrasts, the ab initio model of the complex cannot be constructed with Dammif. Here, one would need to use Monsa. The use of ab initio modelling is also missing from Material and Methods. (as this model does not add to much "more value" it could also be omitted)

* what is the fit of just the GraT-GraA structure (with the missing 22 residues) to the SAXS data?

* I would stay moderate in the interpretation of the dimensionless Kratky plot

Line 265

... dimensionless Kratky plot (Figure 5C) shows a slight disorder with a small shift of the maximum towards higher values....

- the plot also looks a bit different than the one from SASBDB (were R_g and $I(0)$ used as listed in table S2?)

Figure S6, would it be sufficient to only show the 4state fit? The other fits seem kind of obsolete, as only the 4 state fit is discussed.

The very narrow distribution of the "4 state fit" in GraT and for very low R_g values, would normally suggest a not too flexible compact structure. also the selected models shown in Figure 5 D, are more elongated; are these really only weighted as 0.2 and the compact models as 0.8; maybe a different color would help see the compact models

◇ I don't quite understand how the "HigB-like N-terminus" was created- Were the rigid parts structurally aligned and then a hybrid model created? Out of curiosity I fit HigB (from 5ifg) to the data and that already gives a better fit then then the "rigid" model used here (χ^2 7.9; and by leaving the first 25 residues away 5.5))

◇ I also used this pdb entry to run EOM (a program like multifoxx, but which I am more familiar with) ◇ the result was similar to multifoxx; a narrow distribution with low R_g and small D_{max} and only 20-30 % very extended (large R_g and D_{max})

◇ the question I just pose, does this mean there is a two-state system switching between compact and rigid or is it an artefact because there are still some larger components that contributed to the scattering.

◇ for example, the $P(r)$ is very untypical for such a small protein ◇ with a D_{max} of 7 nm alone ◇ then the complex with two GraA and one more GraT would only be 0.9nm larger?

In conclusion:

I understand the attempt to explain the intriguing observation that the N-terminus on the one hand is crucial for the function but on the other hand is not visible in the x-tal structure

◇ and yes, the statement that this is so because of flexibility is a valid hypothesis

◇ however, I would treat it as such (hypothesis) and state it more carefully

For one, perhaps in Figure 5, not use the "fixed N-terminus" but compare it to the x-tal structure with the missing residues

◇ also in Figure S8 ◇ it seems kind of like 'cherry picking' to say that the hypothetical complex can't form, and then use a "random but rather extended" confirmation (even though according to Figure S6A ◇ the distribution suggests it is compact)

◇ and (as already mentioned) ◇ the title could be revised to set a different focus.

REVIEWERS' COMMENTS:

Reviewer #2 (Remarks to the Author):

The Authors have gone to quite some length to try to collect further evidence that the N-terminus of GraT is indeed flexible. I still do not quite understand why this point has to be emphasized so strongly. As also suggested by Referee 1, the focus (of eg. the title) could be more on the promotor binding instead. All the other data is nicely presented, the MS well written and an interesting topic. As stated below I would suggest down playing the “flexibility”.

We understood that the referees were not fully convinced about the presence of an IDP region in the toxin, especially in its free state. We therefore wanted to gather as much data as possible about the flexibility of GraT to explain its ability in down-regulating the gene repressor activity of GraA when it is bound the *graTA* operator.

- The SAXS data have been made available upon request (also corresponding tables have been added to the MS and the data uploaded to the SASBDB database)

*** it would be nice if the models could later be added to database as well.**

*** the section on Kratky plot (lines 230 -239) could be placed in Material and Methods**

We will later deposit the models derived from the different SAXS curves. Following the suggestions we moved the section about the Kratky plot to the materials and methods.

Some questions arise regarding the SAXS data:

*** SEC-SAXS data collection has been performed. It would be good to state how $I(0)$ values were derived from here, ie. how was the concentration measured and taken into account? Or were other concentration independent MW estimates made.**

***Note, R_g is derived from the Guinier Plot (which also delivers $I(0)$), please correct: “ R_g is derived from $I(0)$ ” in the material and method section**

This section now reads: “Each protein was ran through a Shodex KW402.5-4F at 0.2 ml/min. Scattering curves covering a concentration range around the peak were normalized and averaged to obtain the final scattering curve. R_g values were derived from the value of I_0 which were obtained by extrapolating to $q = 0$ using the Guinier approximation as implemented in ATSAS suite⁶⁴. The molecular weight of the different entities were estimated in a concentration independent way using the I_0 , Porod volume and Fisher methods.”

As DNA and protein have different contrasts, the ab initio model of the complex cannot be constructed with Dammif. Here, one would need to use Monsa. The use of ab initio modelling is also missing from Material and Methods. (as this model does not add to much “more value” it could also be omitted)

We agree with the referee and decided to remove this part from the paper as it is not crucial.

What is the fit of just the GraT-GraA structure (with the missing 22 residues) to the SAXS data?

Using the online program FOXS (the same we used for all the other analysis) it gives a $\chi^2=10.8$.

I would stay moderate in the interpretation of the dimensionless Kratky plot. The plot also looks a bit different than the one from SASBDB (were R_g and $I(0)$ used as listed in table S2?)

Following the suggestion of the reviewer we changed the text and reads like: “...the dimensionless Kratky plot (Figure 5C) shows a slight disorder with a small shift of the maximum towards higher values (2.0; 1.22)...”

Yes, they are the same values: $R_g = 16 \text{ \AA}$ and $I_0 = 0.01444$

Figure S6, would it be sufficient to only show the 4state fit? The other fits seem kind of obsolete, as only the 4 state fit is discussed.

Based on this comment we remade the figure. Now, it only shows the four-states model (in red) and the complete ensemble (in black).

The very narrow distribution of the "4 state fit" in GraT and for very low RG values, would normally suggest a not too flexible compact structure. also the selected models shown in Figure 5 D, are more elongated; are these really only weighted as 0.2 and the compact models as 0.8; maybe a different color would help see the compact models.

The result of the program Multifoxs consists of eight groups, each of them containing a four states model (Response Figure 1). One model can be present in more than one group. For example, model 1 is present in all the groups, making the highest weighted conformation. The total weight of the compact conformations ($\langle R_G \rangle = 14.3 \text{ \AA}$) is determined by taking into account the multiplicity of the different compact models, while the more extended conformations ($19 \text{ \AA} < R_G < 24$) have a weight of ~ 0.2 .

This high multiplicity is what makes the compact conformations look less abundant than the extended ones. Nevertheless, this structure can still be rather dynamic with a probable vast conformational space ranging from model 1 to model 2 (Response Figure 1).

Response Figure 1. It shows the eight groups of GraT 4-states models that best fit the SAXS curve. The table displays the model number, as marked in the left, with their respective R_G s, and in parenthesis the population size of each model.

I don't quite understand how the "HigB-like N-terminus" was created- Were the rigid parts structurally aligned and then a hybrid model created?

To create this model the residues 25 to 92 of GraT were aligned with the equivalent residues in the amino acid sequence of HigB from *Proteus vulgaris* (PDB code 4MCX). Then we used the program Modeller to generate the GraT N-terminus from residue 1 to 24 using HigB from *Proteus vulgaris* as template.

Out of curiosity I fit HigB (from 5ifg) to the data and that already gives a better fit than the “rigid” model used here (chi2 7.9; and by leaving the first 25 residues away 5.5))

PDB entry 5ifg corresponds to *E. coli* HigB, which again has a similar structure but is of course not identical to *Pseudomonas* HigB. This explains the different chi2 value (while still quite high).

Repeating the same experiment with HigB from PDB code 5ifg, but in this case using the program FOXS results in $\text{Chi}^2 = 13.6$ for the full length protein. With the first 25 amino acids removed the fitting gets worse, with $\text{Chi}^2 = 19.9$.

I also used this pdb entry to run EOM (a program like multifoys, but which I am more familiar with) the result was similar to multifoys; a narrow distribution with low RG and small Dmax and only 20-30 % very extended (large Rg and Dmax) the question I just pose, does this mean there is a two-state system switching between compact and rigid or is it an artifact because there are still some larger components that contributed to the scattering: for example, the P(r) is very untypical for such a small protein with a Dmax of 7 nm alone then the complex with two GraA and one more GraT would only be 0.9nm larger?

We give the reason to the referee that the reported Dmax for GraT is very high for a protein of 98 amino acids. This was my mistake (the first author). I used a wrong P(r) function file to complete the table and to upload it to SASDB database. My apologies for that. From the correct P(r) function, obtained with GNOM with default parameters, the Dmax is 54.6 Å. This value is now in agreement with the size of a monomer of GraT. This issue is already fixed in the text and the SASDB database (still needs to be done). Furthermore, the elution volume in the SEC experiment, carried out just before SAXS, also shows that GraT is monomer of ~ 10 kDa as shown in the response figure 2.

Response figure 2: SEC carried out with GraT. The black curve is the HPCL chromatogram of GraT just before the SAXS. The red line is the calibration curve using the Biorad gel filtration standards represented by the squares.

In conclusion:

I understand the attempt to explain the intriguing observation that the N-terminus on the one hand is crucial for the function but on the other hand is not visible in the x-tal structure and yes, the statement that this is so because of flexibility is a valid hypothesis however, I would treat it as such (hypothesis) and state it more carefully.

Form the SAXS data we can say that GraT exist in an equilibrium between an extended (20-30%) and a more compact conformation (70-80%). Nevertheless, this compact conformation is more extended than

Proteus vulgaris HigB and very likely not unique. This difference in behavior between GraT and HigB stems from the first 22 amino acids.

Another stronger biological data supporting the flexibility of GraT N-terminus is that, despite being bound at the opposite site of the GraA DNA binding region GraT is still able to completely abolish the capacity of the Gra/GraT complex to bind its operator. On the other hand, when the first 22 amino acids of GraT are removed the delta22-GraT /GraA complex binds its operator even stronger than GraA alone.

Combined our data show that (1) the N-terminus of GraA is flexible and (2) that this N-terminus is required for correct functionality. This makes it more than a pure hypothesis since it is backed-up with a significant experimental data. In the end, all conclusions drawn from scientific experiments are hypotheses that can always be further strengthened or eventually be replaced by an even stronger hypothesis. We do not feel, however, that we have left open obvious loose ends that can be confirmed or denied using a simple additional experiment.

For one, perhaps in Figure 5, not use the “fixed N-terminus” but compare it to the x-tal structure with the missing residues also in Figure S8 it seems kind of like ‘cherry picking’ to say that the hypothetical complex can’t form, and then use a “random but rather extended” conformation (even though according to Figure S6A the distribution suggests it is compact) and (as already mentioned) the title could be revised to set a different focus.

We used *P. vulgaris* HigB and its corresponding N-terminus because it is the closest homologue for GraT of all proteins present in the PDB both in terms of structure and in terms of sequence. It is also the best studied one (and the only one for which there is a structure in complex with a ribosome) and the one which was first discovered and used as the paradigm for HigB. For these reasons it is the most obvious choice to start comparisons. Other HigB's such as *V. cholerae* or *E. coli* belong to other classes of the higBA family tree. They do display significant differences, not only in their ordered N-termini, but around the entire molecule.

The idea we want to transmit with Figure S8 is that there are possible extended conformations of GraT that hinder the binding of GraA/GraT complex to DNA. We have shown that GraA binds DNA only when two GraA dimers are bound at the same time, this makes that having at least one GraT in the extended conformation can actually impede the formation of the GraA/GraT/DNA complex.

We would prefer not to change the title as it transmits, in our opinion, the most important finding of this work: the functional importance of an N-terminal IDP region in GraT. This we think is what makes the *graTA* module distinct from other TA modules including closely related ones.